# Near-optimal Distributional Reinforcement Learning towards Risk-sensitive Control

## Abstract

We consider finite episodic Markov decision processes aiming at the entropic risk measure (EntRM) of return for risk-sensitive control. We identify two properties of the EntRM that enable risk-sensitive distributional dynamic programming. We propose two novel distributional reinforcement learning (DRL) algorithms, including a model-free one and a model-based one, that implement optimism through two different schemes. We prove that both of them attain $\tilde{\mathcal{O}}(\frac{\exp(|\beta|H)-1}{|\beta|H}H\sqrt{HS^2AT})$ regret upper bound, where $S$ is the number of states, $A$ the number of states, $H$ the time horizon and $T$ the number of total time steps. It matches RSVI2 proposed in [22] with a much simpler regret analysis. To the best of our knowledge, this is the first regret analysis of DRL, which theoretically verifies the efficacy of DRL for risk-sensitive control. Finally, we improve the existing lower bound by proving a tighter bound of $\Omega(\frac{\exp(\beta H/6)-1}{\beta H}H\sqrt{SAT})$ for $\beta > 0$ case, which recovers the tight lower bound $\Omega(H\sqrt{SAT})$ in the risk-neutral setting.

## 1 Introduction

Standard reinforcement learning (RL) [45] seeks to find an optimal policy that maximizes the expectation of return. It is also called risk-neutral RL since the objective is the mean functional of the return distribution. However, in some high-stakes applications including finance [15, 6], medical treatment [21] and operations [16] etc, the decision-maker tends to be risk-sensitive with the goal of maximizing some risk measure of return distribution.

In this paper, we consider the problem of optimizing the exponential risk measure (EntRM) in the episodic and finite MDP setting for risk-sensitive control. The entropic risk measure can trade-off between the expectation and the variance, and adjusts the risk-sensitiveness by control a risk parameter (see Equation 1). Ever since the seminal work of [29], risk-sensitive RL based on the EntRM has been applied across a wide range of domains [43, 37, 27]. Most of the existing approaches, however, involve complicated algorithmic design to deal with the non-linearity of the EntRM.

Distributional reinforcement learning (DRL) [4] has demonstrated its superior performance over traditional methods in some difficult tasks [14, 13] under risk-neutral setting. Different from the value-based approaches, it learns the whole return distribution instead of a real-valued value function. Given the entire return distribution, it is natural to leverage the distributional information to optimize a risk measure other than expectation [13, 44, 33]. Despite of the intrinsic connection between DRL and risk-sensitive RL, it is surprising that existing works on risk-sensitive control via DRL approaches ([13, 34, 1]) lack regret analysis. Consequently, it is challenging to evaluate and improve these DRL algorithms in terms of sample-efficiency, which brings about a reasonable question

*Can distributional reinforcement learning attain near-optimal regret for risk-sensitive control?*

Submitted to 36th Conference on Neural Information Processing Systems (NeurIPS 2022). Do not distribute.

In this work, we answer this question positively by providing two DRL algorithms with provably regret guarantees. We devise two novel DRL algorithms with principled exploration schemes for risk-sensitive control in the tabular MDP setting. In particular, the proposed algorithms implement the principle of optimism in the face of uncertainty (OFU) at the distributional level to balance the exploration-exploitation trade-off. By providing the first regret analysis of DRL, we theoretically verifies the efficacy of DRL for risk-sensitive control. Therefore, our work bridge the gap between DRL and risk-sensitive RL with regard to sample complexity.

**Main contributions.** We summarize our main contributions in the following.
**1.** We build a risk-sensitive distributional dynamic programming (RS-DDP) framework. To be more specific, we choose the entropic risk measure (EntRM) of the return distribution as our objective. By identifying two key properties of EntRM, We establish distributional Bellman optimality equation for risk-sensitive control.
**2.** We propose two DRL algorithms that enforce the OFU principle in a distributional fashion through two different schemes. We provide $\tilde{\mathcal{O}}(\frac{\exp(|\beta|H)-1}{|\beta|}H\sqrt{S^2AK})$ regret upper bound, which matches the best existing result of RSVI2 in [22]. It is the first regret analysis of DRL algorithm in the finite episodic MDP in the risk-sensitive setting. Compared to [22], our algorithm does not involve complicated bonus design, and our analysis are conceptually cleaner and easier to interpret.
**3.** We fill the gaps in the proof of lower bound in [23]. To the best of our knowledge, [23] only implies a lower bound $\Omega(\frac{\exp(|\beta|H/2)-1}{|\beta|}\sqrt{K})$ rather the claimed bound $\Omega(\frac{\exp(|\beta|H/2)-1}{|\beta|}\sqrt{T})$. The resulting lower bound is independent of $S$ and $A$ and is loose with a factor of $\sqrt{H}$. We overcome these issues by proving a tight lower bound of $\Omega(\frac{\exp(\beta H/6)-1}{\beta H}H\sqrt{SAT})$ for $\beta > 0$. Note that the lower bound is tight in the risk-neutral setting ($\beta \to 0$).

**Related work.** Following the paper [4], DRL has witnessed a rapid growth of study in literature [40, 14, 13, 2, 32]. Most of these works focus on improving the performance in the risk-neutral setting, with a few exceptions [13, 34, 1]. However, none of these works study the sample complexity.

A rich body of work studies risk-sensitive RL with the EntRM [7, 8, 10, 9, 3, 11, 12, 18, 17, 19, 24, 28, 30, 33, 35, 36, 38, 39, 42, 43]. In particular, [29] is the first to introduce the ERM as risk-sensitive objective in MDP. However, they either assume known transition and reward or consider infinite-horizon setting without sample-complexity considerations.

Two works are closely related to ours [23, 22] under precisely the same setting. [23] is the first to study the risk-sensitive episodic MDP, which provides the first algorithms and regret guarantees. Nevertheless, the regret upper bounds contain a dispensable factor of $\exp(|\beta|H^2)$. Additionally, their lower bound proof contains mistakes, and the corrected proof suggests a weaker bound. [22] improves the algorithm by removing the additional $\mathcal{O}(\exp(|\beta|H^2))$ factor. However, the regret analysis is complicated, and the lower bound is not fixed. A very recent work ([1]) independently proposes a risk-sensitive DDP framework, but their work is fundamentally different from ours. The risk measure considered in [1] is the conditional value at risk (CVaR), and they focus on the infinite horizon setting. Due to the space limit, we provide detailed comparisons with [23, 22, 1] in Appendix A.

## 2 Preliminaries

**Notations.** We write $[M:N] \triangleq \{M, M+1, ..., N\}$ and $[N] \triangleq [1:N]$ for any positive integers $M \leq N$. We adopt the convention that $\sum_{i=n}^{m} a_i \triangleq 0$ if $n > m$ and $\prod_{i=n}^{m} a_i \triangleq 1$ if $n > m$. We use $\mathbb{I}\{\cdot\}$ to denote the indicator function. For any $x \in \mathbb{R}$, we define $[x]^+ \triangleq \max\{x, 0\}$. We define the step function with parameter $c$ as $\psi_c(x) \triangleq \mathbb{I}\{x \geq c\}$. Note that $\psi_c$ represents the CDF of a deterministic variable taking value $c$. We denote by $\mathscr{D}([a,b])$, $\mathscr{D}_M$ and $\mathscr{D}$ the set of distributions supported on $[a, b]$, $[0, M]$ and the set of all distributions respectively. For a random variable (r.v.) $X$, we use $\mathbb{E}[X]$ and $\mathbb{V}[X]$ to denote its expectation and variance. For two r.v.s, we denote by $X \perp Y$ if $X$ is independent of $Y$. We use $\tilde{\mathcal{O}}(\cdot)$ to denote $\mathcal{O}(\cdot)$ omitting logarithmic factors.

**Episodic MDP.** An episodic MDP is identified by $\mathcal{M} \triangleq (\mathcal{S}, \mathcal{A}, (P_h)_{h\in[H]}, (\mathcal{R}_h)_{h\in[H]}, H)$, where $\mathcal{S}$ is the state space, $\mathcal{A}$ the action space, $P_h : \mathcal{S} \times \mathcal{A} \times \to \Delta(\mathcal{S})$ the probability transition kernel at step $h$, $\mathcal{R}_h : \mathcal{S} \times \mathcal{A} \to \mathscr{D}([0,1])$ the collection of reward distributions at step $h$ and $H$ the length of

85 one episode. The agent interacts with the environment for $K$ episodes. At the beginning of episode $k$,
86 Nature selects an initial state $s_1^k$ arbitrarily. In step $h$, the agent takes action $a_h^k$ and observes random
87 reward $R_h^k(s_h^k, a_h^k) \sim \mathcal{R}_h(s_h^k, a_h^k)$ and reaches the next state $s_{h+1}^k \sim P_h(\cdot|s_h^k, a_h^k)$. The episode
88 terminates at $H + 1$ with $R_{H+1}^k = 0$, then the agent proceeds to next episode.

89 For each $(k, h) \in [K] \times [H]$, we denote by $\mathcal{H}_h^k \triangleq \left(s_1^1, a_1^1, s_2^1, a_2^1, \ldots, s_H^1, a_H^1, \ldots, s_h^k, a_h^k\right)$ the
90 (random) history up to step $h$ episode $k$. We define $\mathcal{F}_k \triangleq \mathcal{H}_H^{k-1}$ as the history up to episode
91 $k - 1$. We describe the interaction between the algorithm and MDP in two levels. In the level of
92 episode, we define an algorithm as a sequence of function $\mathscr{A} \triangleq (\mathscr{A}_k)_{k \in [K]}$, each mapping $\mathcal{F}_k$ to
93 a policy $\mathscr{A}_k(\mathcal{F}_k) \in \Pi$. We denote by $\pi^k \triangleq \mathscr{A}_k(\mathcal{F}_k)$ the policy at episode $k$. In the level of step, a
94 (deterministic) policy $\pi$ is defined as a sequence of functions $\pi = (\pi_h)_{h \in [H]}$ with $\pi_h : \mathcal{S} \to \Delta(\mathcal{A})$.

95 **Entropic risk measure.** EntRM is a well-known risk measure in risk-sensitive decision-making,
96 including mathematical finance [25], Markovian decision processes [3]. The EntRM value of a r.v.
97 $X \sim F$ with coefficient $\beta \neq 0$ is defined as

$$U_\beta(X) \triangleq \frac{1}{\beta} \log(\mathbb{E}_{X \sim F}[\exp(\beta X)]) = \frac{1}{\beta} \log\left(\int_{\mathbb{R}} \exp(\beta x) dF(x)\right).$$

98 With slight abuse of notations, we write $U_\beta(F) = U_\beta(X)$ for $X \sim F$. For $\beta$ with small absolute
99 value, using Taylor's expansion we have

$$U_\beta(X) = \mathbb{E}[X] + \frac{\beta}{2}\mathbb{V}[X] + \mathcal{O}(\beta^2). \tag{1}$$

100 Hence for a decision-maker who aims at maximizing the EntRM value, she tends to be risk-seeking
101 (favoring high uncertainty in $X$) if $\beta > 0$ and risk-averse (favoring low uncertainty in $X$) if $\beta < 0$.
102 $|\beta|$ controls the risk-sensitivity. It exactly recovers mean as the risk-neutral objective when $\beta \to 0$.

## 3 Risk-sensitive Distributional Dynamic Programming

104 [4, 40] has discussed the *infinite-horizon* distributional dynamic programming in the *risk-neutral*
105 setting, which will be referred to as the classical DDP. There is a big gap between the risk-sensitive
106 MDP and the risk-neutral one. In this section, we establish the novel DDP framework for risk-sensitive
107 control.

108 We start with defining the return for a policy $\pi$ starting from state-action pair $(s, a)$ at step $h$

$$Z_h^\pi(s, a) \triangleq \sum_{h'=h}^{H} R_{h'}(s_{h'}, a_{h'}), \ s_h = s, a_{h'} = \pi_{h'}(s_{h'}), s_{h'+1} \sim P_{h'}(\cdot|s_{h'}, a_{h'}).$$

109 Define $Y_h^\pi(s) \triangleq Z_h^\pi(s, \pi_h(s))$. There are three sources of randomness in $Z_h^\pi(s, a)$: the reward
110 $R_h(s, a)$, the transition $P^\pi$ and the next-state return $Y_{h+1}^\pi(s_{h+1})$. Denote by $\nu_h^\pi(s)$ and $\eta_h^\pi(s, a)$ the
111 cumulative distribution function (CDF) corresponding to $Y_h^\pi(s)$ and $Z_h^\pi(s, a)$ respectively. To the
112 end of risk-sensitive control, we define the action-value function of a policy $\pi$ at step $h$ as $Q_h^\pi(s, a) \triangleq$
113 $U_\beta(Z_h^\pi(s, a))$, i.e. the EntRM value of the return distribution, for each $(s, a, h) \in \mathcal{S} \times \mathcal{A} \times [H]$. The
114 value function is defined as $V_h^\pi(s) \triangleq Q_h^\pi(s, \pi_h(s)) = U_\beta(Y_h^\pi(s))$.

115 We focus on the control setting, in which the goal is to find an optimal policy to maximize the value
116 function, i.e.

$$\pi^* \triangleq \arg \max_{(\pi_1, \ldots, \pi_H) \in \Pi} V_1^{\pi_1 \cdots \pi_H}(s).$$

117 We write $\pi = (\pi_1, ..., \pi_H)$ to emphasize that it is a multi-stage maximization problem. Direct search
118 suffers exponential computational complexity. In the risk-neutral case, the *principle of optimality*
119 holds, i.e.,the optimal policy of tail sub-problem is the tail optimal policy [5]. Therein the multi-stage
120 maximization problem can be reduced to a multiple single-stage maximization problem. However,
121 the principle does not always hold for general risk measures. For example, the optimal policy for
122 CVaR may be non-Markovian/history-dependent ([41]).

123 We identify two key properties of EntRM, upon which we retain the principle of optimality.

124 **Lemma 1.** *The EntRM satisfies the following properties:*

- *Additive:* $X \perp Y \Rightarrow U_\beta(X + Y) = U_\beta(X) + U_\beta(Y), \ \forall X, Y.$

- *Monotonicity-preserving:* $\forall F_1, F_2, G \in \mathscr{D}, \ \forall \theta \in [0, 1],$

$$U_\beta(F_2) \leq U_\beta(F_1) \Rightarrow U_\beta((1-\theta)F_2 + \theta G) \leq U_\beta((1-\theta)F_1 + \theta G).$$

127 The proof is given in Appendix B. In particular, the additivity entails that the EntRM value of the
128 current return $Z_h^\pi(s, a)$ equals the sum of the immediate value of $R_h(s, a)$ and the value of the future
129 return $Y_h^\pi(s')$, i.e.,

$$U_\beta(Z_h^\pi(s, a)) = U_\beta(R_h(s, a)) + U_\beta(Y_h^\pi(s')).$$

130 The monotonicity-preserving property together with the additivity suggests that the optimal future
131 return $Y_h^*(s')$ consists in the optimal current return $Z_h^*(s, a)$

$$Z_h^*(s, a) = R_h(s, a) + Y_h^*(s').$$

132 These observations implies the principle of optimality.

133 **Proposition 1** (Principle of optimality). *Let $\pi^* = \{\pi_1^*, \pi_2^*, ..., \pi_H^*\}$ be an optimal policy and assume*
134 *when we visit some state $s$ using policy $\pi$ at time-step $h$ with positive probability. Consider the*
135 *sub-problem defined by the the following maximization problem*

$$\max_{\pi \in \Pi} V_h^\pi(s) = U_\beta(\mathcal{R}_h(s, a)) + U_\beta([P_h \nu_{h+1}^\pi](s, a)).$$

136 *Then the truncated optimal policy $\{\pi_h^*, \pi_{h+1}^*, ..., \pi_H^*\}$ is optimal for this sub-problem.*

137 The proof is given in Appendix E. It further induces the distributional Bellman optimality equation.

138 **Proposition 2** (Distributional Bellman optimality equation). *For arbitrary initial state $s_1$, the optimal*
139 *policy $(\pi_h^*)_{h\in[H]}$ is given by the following backward recursions:*

$$\nu_{H+1}^*(s) = \psi_0, \ \eta_h^*(s, a) = [P_h \nu_{h+1}^*](s, a) * f_h(\cdot | s, a),$$
$$\pi_h^*(s) = \arg \max_{a \in \mathcal{A}} Q_h^*(s, a) = U_\beta(\eta_h^*(s, a)), \nu_h^*(s) = \eta_h^*(s, \pi_h^*(s)), \tag{2}$$

140 *where $f_h(s, a)$ is the probability density function of $R_h(s, a)$. Furthermore, the sequence $(\eta_h^*)_{h\in[H]}$*
141 *and $(\nu_h^*)_{h\in[H]}$ are the sequence of distributions corresponding to the optimal returns at each step.*

142 The proof is given in Appendix E. For simplicity, we define the distributional Bellman operator
143 $\mathcal{B}(P, \mathcal{R}) : \mathscr{D}^\mathcal{S} \to \mathscr{D}^{\mathcal{S} \times \mathcal{A}}$ with associated model $(P, \mathcal{R}) = (P(s, a), \mathcal{R}(s, a))_{(s,a)\in\mathcal{S}\times\mathcal{A}}$ as

$$[\mathcal{B}(P, \mathcal{R})\nu](s, a) \triangleq [P\nu](s, a) * f_h(\cdot | s, a), \ \forall(s, a) \in \mathcal{S} \times \mathcal{A}.$$

144 Hence we can rewrite Equation 2 in a compact form:

$$\nu_{H+1}^*(s) = \psi_0, \ \eta_h^*(s, a) = [\mathcal{B}(P_h, \mathcal{R}_h)\nu_{h+1}^*](s, a),$$
$$\pi_h^*(s) = \arg \max_{a \in \mathcal{A}} U_\beta(\eta_h^*(s, a)), \ \nu_h^*(s) = \eta_h^*(s, \pi_h^*(s)), \forall(s, a, h) \in \mathcal{S} \times \mathcal{A} \times [H]. \tag{3}$$

145 Finally, we define the regret of an algorithm $\mathscr{A}$ interacting with an MDP $\mathcal{M}$ for $K$ episodes as

$$\text{Regret}(\mathscr{A}, \mathcal{M}, K) \triangleq \sum_{k=1}^{K} V_1^*(s_1^k) - V_h^{\pi^k}(s_1^k).$$

146 Note that the regret is a random variable since $\pi^k$ is a random quantity. We denote by
147 $\mathbb{E}[\text{Regret}(\mathscr{A}, \mathcal{M}, K)]$ the expected regret. We will omit $\pi$ and $\mathcal{M}$ if it is clear from the context.

## 148 4 Algorithm

149 For a better understanding of the readers, we present our algorithms under the assumption that
150 the reward is *deterministic and known*[1]. The algorithms for the case of random reward are given

---

[1]The algorithms for random reward enjoy the regret bounds of the same order.

in Appendix C. We denote by $\{r_h(s,a)\}_{(s,a,h)\in\mathcal{S}\times\mathcal{A}\times[H]}$ the reward functions. For the case of deterministic reward, the Bellman update in Equation 2 takes the form

$$\eta_h^*(s,a) = [P_h\nu_{h+1}^*](s,a)(\cdot - r_h(s,a)),$$

since adding a deterministic reward $r_h(s,a)$ corresponds to shifting the distribution $[P_h\nu_{h+1}^*](s,a)$ by an amount of $r_h(s,a)$. We thus define the distributional Bellman operator $\mathcal{B}(P,\mathcal{R}) : \mathscr{D}^{\mathcal{S}} \to \mathscr{D}^{\mathcal{S}\times\mathcal{A}}$ with associated model $(P,r) = (P(s,a), r(s,a))_{(s,a)\in\mathcal{S}\times\mathcal{A}}$ as

$$[\mathcal{B}(P,r)\nu](s,a) \triangleq [P\nu](s,a)(\cdot - r_h(s,a)), \ \forall(s,a) \in \mathcal{S} \times \mathcal{A}.$$

We propose two DRL algorithms in this section, including a model-free algorithm and a model-based algorithm. We first introduce the **M**odel- **F**ree **R**isk-sensitive **O**ptimistic **D**istribution **I**teration (RODI-MF) in Algorithm 1. For completeness, we introduce some additional notations here. For two CDFs $F$ and $G$ over reals, we define the supremum distance between them $\|F - G\|_\infty \triangleq \sup_x |F(x) - G(x)|$. We define the $\ell_1$ distance between two probability mass functions (PMFs) $P$ and $Q$ as $\|P - Q\|_1 \triangleq \sum_i |P_i - Q_i|$. We denote by $B_\infty(F,c) := \{G \in \mathscr{D} | \|G - F\|_\infty \leq c\}$ the supremum norm ball centered at $F$ with radius $c$. With slight abuse of notations, we denote by $B_1(P,c)$ the $l_1$ norm ball centered at $P$ with radius $c$.

## 4.1 Algorithm overview

### 4.1.1 RODI-MF

In each episode, the algorithm includes the planning phase (Line 4-12) and the interaction phase (Line 13-17).

**Planning phase.** In a high level, the algorithm implements an optimistic version of approximate DDP from step $H + 1$ to step 1 in each episode. In Line (5-7), it performs sample-based Bellman update. To make it clear, we introduce the superscript $k$ to the variables of Algorithm 1 in episode $k$. For example, $\eta_h^k$ denotes $\eta_h$ in episode $k$. Specifically, for those visited state-action pairs, we claim that Line 6 is equivalent to a model-based Bellman update. Denote by $\mathbb{I}_h^k(s,a) \triangleq \mathbb{I}\{(s_h^k, a_h^k) = (s,a)\}$. Fix a tuple $(s,a,k,h)$ such that $N_h^k(s,a) \geq 1$. We denote by $\hat{P}_h^k(\cdot|s,a)$ the empirical transition model

$$\hat{P}_h^k(s'|s,a) = \frac{1}{N_h^k(s,a)} \sum_{\tau\in[k-1]} \mathbb{I}_h^\tau(s,a) \cdot \mathbb{I}\{s_{h+1}^\tau = s'\}.$$

Observe that for any $\nu \in \mathscr{D}^{\mathcal{S}}$, we have

$$\left[\hat{P}_h^k\nu\right](s,a) = \sum_{s'\in\mathcal{S}} \hat{P}_h^k(s'|s,a)\nu(s') = \frac{1}{N_h^k(s,a)} \sum_{s'\in\mathcal{S}} \sum_{\tau\in[k-1]} \mathbb{I}_h^\tau(s,a) \cdot \mathbb{I}\{s_{h+1}^\tau = s'\}\nu(s')$$

$$= \frac{1}{N_h^k(s,a)} \sum_{\tau\in[k-1]} \mathbb{I}_h^\tau(s,a) \cdot \sum_{s'\in\mathcal{S}} \mathbb{I}\{s_{h+1}^\tau = s'\}\nu(s_{h+1}^\tau)$$

$$= \frac{1}{N_h^k(s,a)} \sum_{\tau\in[k-1]} \mathbb{I}_h^\tau(s,a)\nu(s_{h+1}^\tau).$$

Hence the update formula in Line 6 of Algorithm 1 can be rewritten as

$$\eta_h^k(s,a) = \left[\hat{P}_h^k\nu_{h+1}^k\right](s,a)(\cdot - r_h(s,a)) = \left[\mathcal{B}(\hat{P}_h^k, r_h)\nu_{h+1}^k\right](s,a),$$

implying the equivalence to a model-based Bellman update with empirical model $\hat{P}_h^k$. Alternatively, the unvisited $(s,a)$ remains to be the return distribution corresponding to the highest possible reward $H + 1 - h$. The algorithm then computes the optimism constants (Line 8) and enforces OFU through the distributional optimism operator $c_h^k$ (Line 9) to obtain the optimistically plausible return distribution $\eta_h^k$. The choice of $c_h^k$ will be discussed later. The optimistic return distributions yields the optimistic value function, from which the algorithm generates the greedy policy $\pi_h^k$. The policy $\pi_h^k$ will be used in the interaction phase.

184 **Interaction phase.** In Line (15-16), the agent interacts with the environment using policy $\pi$ and
185 updates the counts $N_h$ based on new observations.

---

**Algorithm 1** RODI-MF

---

1: Input: $T$ and $\delta$
2: Initialize $N_h(\cdot,\cdot) \leftarrow 0$; $\eta_h(\cdot,\cdot), \nu_h(\cdot) \leftarrow \psi_{H+1-h}$ for all $h \in [H]$
3: **for** $k = 1 : K$ **do**
4:     **for** $h = H : 1$ **do**
5:         **if** $N_h(\cdot,\cdot) > 0$ **then**
6:             $\eta_h(\cdot,\cdot) \leftarrow \frac{1}{N_h(\cdot,\cdot)} \sum_{\tau \in [k-1]} \mathbb{I}_h^\tau(\cdot,\cdot) \nu_{h+1}(s_{h+1}^\tau)(\cdot - r_h(\cdot,\cdot))$
7:         **end if**
8:         $c_h(\cdot,\cdot) \leftarrow \sqrt{\frac{2S}{N_h(\cdot,\cdot) \vee 1} \iota}$
9:         $\eta_h(\cdot,\cdot) \leftarrow \mathrm{O}_{c_h(\cdot,\cdot)}^\infty \eta_h(\cdot,\cdot)$
10:        $\pi_h(\cdot) \leftarrow \arg\max_a U_\beta(\eta_h(\cdot,a))$
11:        $\nu_h(\cdot) \leftarrow \eta_h(\cdot, \pi_h(\cdot))$
12:    **end for**
13:    Receive $s_1^k$
14:    **for** $h = 1 : H$ **do**
15:        $a_h^k \leftarrow \pi_h(s_h^k)$ and transit to $s_{h+1}^k$
16:        $N_h(s_h^k, a_h^k) \leftarrow N_h(s_h^k, a_h^k) + 1$
17:    **end for**
18: **end for**

---

### 4.1.2 RODI-MB

187 We introduce the second algorithm **M**odel- **B**ased **R**isk-sensitive **O**ptimistic **D**istribution **I**teration
188 (RODI-MB). Algorithm 2 is a model-based algorithm because it requires to explicitly maintaining the
189 empirical transition model in each episode. However, it can be reduced to a *non-distributional* rein-
190 forcement learning algorithm that deals with the one-dimensional values instead of the distributions,
191 which saves the computational complexity and space complexity. Likewise, the algorithm includes
192 the planning phase (Line 4-10) and the interaction phase (Line 11-15).

193 **Planning phase.** Analogous to Algorithm 1, the algorithm also performs approximate DDP together
194 with the OFU principle. First, it applies the distributional optimistic operator to the empirical transition
195 model $\hat{P}_h^k$ to get the optimistic transition model $\tilde{P}_h^k$. Then the algorithm uses $\tilde{P}_h^k$ to execute Bellman
196 update to generate the optimistic return distributions $\eta_h^k$. The remaining steps are the same as
197 Algorithm 1.

198 **Interaction phase.** In Line (13-14), the agent interacts with the environment using policy $\pi^k$ and
199 updates the counts $N_h^{k+1}$ and empirical transition model $\hat{P}_h^{k+1}$ based on the new observations.

---

**Algorithm 2** RODI-MB

---

1: Input: $T$ and $\delta$
2: $N_h^1(\cdot,\cdot) \leftarrow 0$; $\hat{P}_h^1(\cdot,\cdot) \leftarrow \frac{1}{S}\mathbf{1}$ for all $h \in [H]$
3: **for** $k = 1 : K$ **do**
4:     $\nu_{H+1}^k(\cdot) \leftarrow \psi_0$
5:     **for** $h = H : 1$ **do**
6:         $\tilde{P}_h^k(\cdot,\cdot) \leftarrow \mathrm{O}_{c_h^k(\cdot,\cdot)}^1 \hat{P}_h^k(\cdot,\cdot)$
7:         $\eta_h^k(\cdot,\cdot) \leftarrow \left[ \mathcal{B}\left(\tilde{P}_h^k, r_h\right) \nu_{h+1}^k \right](\cdot,\cdot)$
8:         $\pi_h^k(\cdot) \leftarrow \arg\max_a U_\beta(\eta_h^k(\cdot,a))$
9:         $\nu_h^k(\cdot) \leftarrow \eta_h^k(\cdot, \pi_h^k(\cdot))$
10:    **end for**
11:    Receive $s_1^k$
12:    **for** $h = 1 : H$ **do**
13:        $a_h^k \leftarrow \pi_h^k(s_h^k)$ and transit to $s_{h+1}^k$
14:        Compute $N_h^{k+1}(\cdot,\cdot)$ and $\hat{P}_h^{k+1}(\cdot,\cdot)$
15:    **end for**
16: **end for**

---

**Algorithm 3** ROVI

---

1: Input: $T$ and $\delta$
2: $N_h^1(\cdot,\cdot) \leftarrow 0$; $\hat{P}_h^1(\cdot,\cdot) \leftarrow \frac{1}{S}\mathbf{1}$ for all $h \in [H]$
3: **for** $k = 1 : K$ **do**
4:     $W_{H+1}^k(\cdot) \leftarrow 1$
5:     **for** $h = H : 1$ **do**
6:         $\tilde{P}_h^k(\cdot,\cdot) \leftarrow \mathrm{O}_{c_h^k(\cdot,\cdot)}^1 \hat{P}_h^k(\cdot,\cdot)$
7:         $J_h^k(\cdot,\cdot) \leftarrow e^{\beta r_h(\cdot,\cdot)} \left[ \tilde{P}_h^k W_{h+1}^k \right](\cdot,\cdot)$
8:         $W_h^k(\cdot) \leftarrow \max_a J_h^k(\cdot,a)$
9:     **end for**
10:    Receive $s_1^k$
11:    **for** $h = 1 : H$ **do**
12:        $a_h^k \leftarrow \arg\max_a J_h^k(s_h^k, a)$ and tran-
           sit to $s_{h+1}^k$
13:        Compute $N_h^{k+1}(\cdot,\cdot)$ and $\hat{P}_h^{k+1}(\cdot,\cdot)$
14:    **end for**
15: **end for**

**Equivalence to `ROVI`.** **R**isk-sensitive **O**ptimistic **V**alue **I**teration (`ROVI`) is a non-distributional algorithm that deals with the real-valued value function rather than the distribution. It is motivated by the *exponential Bellman equation* proposed by [22]. We define the functional exponential EntRM (EERM) $E_\beta$ as the EntRM after the exponential transformation

$$E_\beta(F) \triangleq \exp(\beta(U_\beta(F))) = \int_{\mathbb{R}} \exp(\beta x) dF(x).$$

Define the exponential value functions $W_h(s) \triangleq E_\beta(\nu_h(s))$ and $J_h(s, a) \triangleq E_\beta(\eta_h(s, a))$ for all $(s, a, h)$s. Applying EERM to Equation 3 yields the exponential Bellman equation

$$\begin{aligned}
J_h^*(s, a) &= \exp(\beta r_h(s, a))[P_h W_{h+1}^*](s, a), \\
W_h^*(s) &= \text{sign}(\beta) \max_a \text{sign}(\beta) J_h^*(s, a), \ W_{H+1}^*(s) = 1.
\end{aligned} \tag{4}$$

To verify the equivalence, it is sufficient to show that $J_h^k$ in Algorithm 3 corresponds to the exponential function of $\eta_h^k$ in Algorithm 2. Observe that $E_\beta$ is linear in $F$, hence it follows that

$$\begin{aligned}
E_\beta(\eta_h^k(s, a)) &= E_\beta \left( \left[ \tilde{P}_h^k \nu_{h+1}^k \right](s, a)(\cdot - r_h(s, a)) \right) = \exp(\beta r_h(s, a)) \cdot \left[ \tilde{P}_h^k E_\beta(\nu_{h+1}^k) \right](s, a) \\
&= \exp(\beta r_h(s, a)) \left[ \tilde{P}_h^k W_{h+1}^k \right](s, a) = J_h^k(s, a).
\end{aligned}$$

The two algorithms generate the policy sequence in the same way, implying that their trajectories $\mathcal{H}_H^K$ follow the same distribution. The formal statement is given in Appendix E.

### 4.2 Distributional Optimism

It is common to add a bonus to the reward to ensure optimism in the risk-neutral setting. Specifically, the bonus is closely related to the level of uncertainty, which is quantified by the concentration inequality. Yet, this type of optimism cannot be adapted to the distributional setup. As one of our technical novelty, the *distributional optimism* is introduced for algorithmic design and regret analysis. In particular, we specify two types of distributional optimism operators, which map a statistically plausible distribution (either the empirical model or the return distribution) to a optimistically plausible distribution. Either of them is applied by Algorithm 2 or Algorithm 1.

**Distributional optimism on the return distribution (in Algorithm 1).** For two CDFs $F$ and $G$, we say that $F$ is more optimistic than $G$ (w.r.t. EntRM) if $U_\beta(F) \geq U_\beta(G)$. This reflects the intuition that the more optimistic distribution should own larger EntRM value. Following [31], we define the distributional optimism operator $O_c^\infty : \mathscr{D}([a, b]) \mapsto \mathscr{D}([a, b])$ with level $c \in (0, 1)$ as

$$(O_c^\infty F)(x) \triangleq [F(x) - c\mathbb{I}_{[a,b)}(x)]^+.$$

The optimistic operator shifts the input $F$ down by at most $c$ over $[a, b]$, and retain the value 1 at $b$. It ensures that $O_c^\infty F$ remains in $\mathscr{D}([a, b])$ and dominates all the other CDFs in $\mathscr{D}([a, b])$ in the sense that $(O_c^\infty F)(x) \leq G(x)$ for any $G \in B_\infty(F, c)$. Since EntRM is monotonic, it holds that

$$U_\beta(O_c^\infty F) \geq U_\beta(G), \ \forall G \in B_\infty(F, c).$$

Hence $O_c^\infty F$ is the most optimistic distribution in the infinity ball $B_\infty(F, c)$. In other words, for any CDF $F$ and $G$ satisfying $\|F - G\|_\infty \leq c$, we have $O_c^\infty G \succeq F$. When specialized to the return distributions, we can apply the distributional optimism operator to the estimated return distribution $\eta_h^k$ (Line 9 of Algorithm 1) with the constant $c_h^k$ to ensure $U_\beta(\eta_h^k(s, a)) \geq U_\beta(\eta_h^*(s, a))$. The constant $c_h^k$ quantifies uncertainty in the model estimation, i.e., $\left\| \hat{P}_h^k(s, a) - P_h(s, a) \right\|_1$.

**Distributional optimism on the model (in Algorithm 2).** Given the model, we consider the optimism among the space of PMFs rather than CDFs. Using the $\ell_1$ concentration inequality [46], we get a concentration bound of the empirical PMF of model: with probability at least $1 - \delta$,

$$\left\| \widehat{P}_h^k(s, a) - P_h(s, a) \right\|_1 \leq c_h^k(s, a) = \sqrt{\frac{2S}{N_h^k(s, a)} \log \frac{1}{\delta}} = \tilde{\mathcal{O}} \left( \sqrt{\frac{2S}{N_h^k(s, a)}} \right).$$

We wish to obtain a optimistic transition model $\tilde{P}_h^k(s, a)$ from the empirical one $\widehat{P}_h^k(s, a)$. To be more specific, the return distribution $\eta_h^k$ computed from $\tilde{P}_h^k(s, a)$ and $\nu_{h+1}^k$ should be more optimistic than

the optimal one $\eta_h^*(s,a)$ with high probability. We thus define the distributional optimism operator $O_c^1 : \mathscr{D}(\mathcal{S}) \mapsto \mathscr{D}(\mathcal{S})$ with level $c$ and future return $\nu \in \mathscr{D}^{\mathcal{S}}$ as

$$O_c^1\left(\widehat{P}(s,a), \nu\right) \triangleq \arg \max_{P \in B_1(\widehat{P}(s,a),c)} U_\beta([P\nu]).$$

The ERM satisfy an interesting property that enables an efficient approach to perform $O_c^1$ (see Appendix B). The following holds by using the induction

$$U_\beta\left(\eta_h^k(s,a)\right) = r_h(s,a) + U_\beta\left(\left[\tilde{P}_h^k \nu_{h+1}^k\right][s,a]\right) \geq r_h(s,a) + U_\beta\left(\left[P_h \nu_{h+1}^k\right][s,a]\right)$$
$$\geq r_h(s,a) + U_\beta\left(\left[P_h \nu_{h+1}^*\right][s,a]\right)$$
$$= U_\beta(\eta_h^*(s,a)),$$

which verify the optimism of $\eta_h^k(s,a)$ over $\eta_h^*(s,a)$.

# 5 Regret Analysis

## 5.1 Regret upper bounds

**Theorem 1** (Regret upper bound of `RODI-MF`)**.** *For any $\delta \in (0,1)$, with probability $1-\delta$, the regret of Algorithm 1 under deterministic reward or Algorithm 4 under random reward is bounded as*

$$Regret(\texttt{RODI-MF}, K) \leq \mathcal{O}\left(\frac{1}{|\beta|} L_H H \sqrt{S^2 A K \log(4SAT/\delta)}\right) = \tilde{\mathcal{O}}\left(\frac{\exp(|\beta|H) - 1}{|\beta|} H \sqrt{S^2 A K}\right).$$

The proof is given in Appendix D.

**Theorem 2** (Regret upper bound of `RODI-MB/ROVI`)**.** *For any $\delta \in (0,1)$, with probability $1-\delta$, the regret of Algorithm 1/Algorithm 3 under deterministic reward or Algorithm 4/Algorithm 6 under random reward is bounded as*

$$Regret(\texttt{RODI-MF}, K) = Regret(\texttt{ROVI}, K) \leq \mathcal{O}(\frac{1}{|\beta|} L_H H \sqrt{S^2 A K \log(4SAT/\delta)})$$
$$= \tilde{\mathcal{O}}\left(\frac{\exp(|\beta|H) - 1}{|\beta|} H \sqrt{S^2 A K}\right).$$

The proof is given in Appendix D. The above results match the best-known results in [22]. In particular, our algorithms attain exponentially improved regret bounds than those of RSVI and RSQ in [23] with a factor of $\exp(|\beta|H^2)$. By choosing $|\beta| = \mathcal{O}(1/H)$, we can eliminate the exponential term and achieve polynomial regret bound akin to the risk-neutral setting.

Compared to the traditional/non-distributional analysis dealing with one-dimensional values, our analysis is distribution-centered, called the *distributional analysis*. The distributional analysis deals with the distributions of the return rather than the risk measure values of the return. For example, it involves the operations of the distributions, the optimism between different distributions, the error caused by estimation of distribution, etc. These distributional aspects fundamentally differ from the traditional analysis that deals with the one-dimensional scalars (value functions). Now we recap the technical novelty of our analysis in the following.

**Lipschitz continuity and linearity.**    We identify two important properties of EERM that establishes the regret upper bounds, including the Lipschitz continuity and linearity. Denote by $L_M$ the Lipschitz constant of the EERM $E_\beta : \mathscr{D}([0,M]) \to \mathbb{R}$ with respect to the infinity norm $\|\cdot\|_\infty$. Lemma 2 provides a *tight* Lipschitz constant of EERM. The Lipschitz constant relates the difference between distributions to the difference measured by their EERM values.

**Lemma 2** (Lipschitz property of EERM)**.** *$E_\beta$ is Lipschitz continuous with respect to the supremum norm over $\mathscr{D}_M$ with $L_M = \exp(|\beta|M) - 1$. Moreover, $L_M$ is tight in terms of both $|\beta|$ and $M$.*

Notice that $\lim_{\beta \to 0} L_M = 0$, which coincides with the fact that $\lim_{\beta \to 0} E_\beta = 1$. The linearity of EERM is a key property that sharpens the regret bounds. In contrast, EntRM is non-linear in the distribution, which could induce a factor of $\exp(|\beta|H)$ when controlling the error propagation across time-steps. It would further lead to a compounding factor of $\exp(|\beta|H^2)$ in the regret bound. In summary, the Lipschitz continuity property enables the regret upper bounds of DRL algorithms, and the linearity tightens the bound.

**Distributional optimism.** Another technical novelty in our analysis is the optimism in the face of uncertainty at the distributional level. The traditional analysis uses the OFU to construct a sequence of optimistic value functions. However, our analysis implements the *distributional optimism* that yields a sequence of optimistic return distributions. In particular, we first define a high probability event, under which the true return distribution concentrates around the estimated one with a certain confidence radius. Then we apply the distributional optimism operator to obtain the optimistically plausible return distribution and the optimistic EntRM value. Hence the regret can be bounded by the surrogate regret, with the optimal EntRM value replaced by

$$\text{Regret}(K) = \sum_{k=1}^{K} \frac{1}{\beta} \log \left( W_1^*(s_1^k) \right) - \frac{1}{\beta} \log \left( W_1^{\pi^k}(s_1^k) \right) \le \frac{1}{\beta} \sum_{k=1}^{K} W_1^k(s_1^k) - W_1^{\pi^k}(s_1^k).$$

**Distributional analysis vs. non-distributional analysis.** When analyzing Algorithm 2/Algorithm 3, proving the regret bound of either algorithm suffices due to their equivalence relation. Since Algorithm 3 is a non-distributional algorithm, one may consider using the standard analysis that does not involve distributions. However, we show that this induces a factor of $\frac{1}{|\beta|} \exp(|\beta|H)$, which explodes as $|\beta| \to 0$. We overcome this issue by invoking a novel distributional analysis of Algorithm 2, leading to the desired factor of $\frac{1}{|\beta|} \left( \exp(|\beta|H) - 1 \right)$.

Although we focus on the algorithms for the deterministic reward in the main text, the regret upper bounds also hold for case of random reward. Algorithm 4, Algorithm 5 and Algorithm 6 corresponds to Algorithm 1, Algorithm 2 and Algorithm 3 respectively (cf. Appendix C).

## 5.2 Regret lower bound

We provide more details of the mistakes in the lower bound of [23] in Appendix D. The proof of [23] reduces the regret lower bound to the two-armed bandit regret lower bound. Since the two-armed bandit is a special case of MDP with $S = 1$, $A = 2$ and $H = 1$, the reduction-based proof only leads to a lower bound independent of $S, A$, and $H$. Instead, our tight lower bound follows a totally different roadmap motivated by [20]. [20] proves the tight minimax lower bound $H\sqrt{SAT}$ for risk-neutral MDP. However, the generalization to risk-sensitive MDP is non-trivial. The main technical challenge is due to the non-linearity of EntRM. The proof in [23] heavily relies on the linearity of expectation, allowing the exchange between taking the risk measure (expectation) and the summation. In the risk-sensitive setting, the non-linearity of EntRM requires new proof techniques.

**Assumption 1.** *Assume $S \ge 6$, $A \ge 2$, and there exists an integer $d$ such that $S = 3 + \frac{A^d - 1}{A - 1}$. We further assume that $H \ge 3d$ and $\bar{H} \triangleq \frac{H}{3} \ge 1$.*

**Theorem 3** (Tighter lower bound). *Assume Assumption 1 holds and $\beta > 0$. Let $\bar{L} \triangleq (1 - \frac{1}{A})(S - 3) + \frac{1}{A}$. Then for any algorithm $\mathscr{A}$, there exists an MDP $\mathcal{M}_{\mathscr{A}}$ such that for $K \ge 2\exp(\beta(H - \bar{H} - d))\bar{H}\bar{L}A$ we have*

$$\mathbb{E}[\text{Regret}(\mathscr{A}, \mathcal{M}_{\mathscr{A}}, K)] \ge \frac{1}{72\sqrt{6}} \frac{\exp(\beta H/6) - 1}{\beta H} H\sqrt{SAT}.$$

The proof is given in Appendix D. Theorem 3 recovers the tight lower bound for standard episodic MDP, implying that the exponential dependence on $|\beta|$ and $H$ in the upper bounds is indispensable. Yet, it is not clear whether a similar lower bound holds for $\beta < 0$, which is left as a future direction.

## 6 Conclusion

We propose a risk-sensitive distributional dynamic programming framework. We devise two novel DRL algorithms, including a model-free one and a model-based one, which implement the OFU principle at the distributional level to balance the exploration and exploitation trade-off under the risk-sensitive setting. We prove that both attain near-optimal regret upper bounds compared with our improved lower bound.

There are several promising future directions. The current regret upper bound has an additional factor $\sqrt{HS}$ compared with the lower bound. It might be possible to remove the factor by designing new algorithms or improving the analysis. Besides, it is interesting to extend the DRL algorithm from tabular MDP to linear function approximation setting. Finally, it will be meaningful to investigate whether the DDP framework holds for other risk measures.

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
