## Negative Social Impact

This script may provide better guidance for risk-sensitive reinforcement learning community. It would have certain negative social impact if the proposed algorithms are deployed for illegal usage.

## A  Comparisons with Related Works

**Comparison with [1]**   We summarize the differences between our work and [1] as follows.

- Setting. [1] considers the discounted MDP with infinite horizon, but we consider the episodic MDP setting. Moreover, [1] assumes that the model is known, while we propose DRL algorithms when the model is unknown (i.e., the learning). Neither RL algorithms suitable for unknown model nor sample complexity guarantee is provided in their work.
- Risk measure. [1] establish the risk-sensitive DDP framework using the risk measure Conditional Value at Risk, while our work considers the entropic risk measure.

**Comparison with [23, 22]**   [21,22] solved the risk-sensitive MDP problem using *valued-based* RL, which estimates and constructs the optimistic version of the (EntRM) value function. [21] proposed the RSVI2 algorithm that improved upon [22] and achieved the best result with the regret upper bound of $\tilde{\mathcal{O}}(\frac{\exp(|\beta|H)-1}{|\beta|}H\sqrt{S^2AK})$. The significance of the proposed algorithms is three-fold.

- Our algorithms are the first distributional reinforcement learning algorithms with provably regret guarantees, suggesting that DRL can work well and even matches the performance of the SOTA value-based RL algorithm for risk-sensitive control in terms of sample complexity. The idea of leveraging the distributional information for risk-sensitivity purposes is natural since the risk measure value is obtained by applying the risk measure/functional to the return distribution. However, existing works on risk-sensitive control via DRL approaches [12, 31, 1] lack regret analysis. Thus, it is difficult to evaluate and improve their algorithms for sample efficiency. Therefore, our algorithms with near-optimal regret upper bounds bridge the gap between the DRL and risk-sensitive MDP in the theoretic RL community.
- Compared with [21], our algorithms are simpler and easier to interpret, leading to clean regret analysis. [21] implements optimism by adding a bonus to the risk measure value function. It designed an exploration mechanism called doubly decaying bonus to remove the $\exp(|\beta|H^2)$ factor from [22]. The doubly decaying bonus decays across the episode and the horizon, which is complicated and not straightforward. Instead, our algorithms implement the distributional optimism by iteratively constructing the optimistic return distribution. The distributional optimism does not involve a complicated bonus design. It only requires a simple application of distributional optimism operator with a constant decaying across the episode. Moreover, the doubly decaying bonus obscures the regret analysis, while our distributional-based analysis is clean and easy to follow.
- Our algorithm may be generalized to risk-sensitive MDP with other risk measures. The analysis of [22,23] is particularly suitable for the EntRM. It is unclear whether it is possible to extend to other risk measures. Under the distributional perspective, our algorithm maintains a sequence of optimistically plausible estimates of the return distribution. Since the distributional information suffices to deal with any risk measure, our algorithm may motivate the design of similar algorithms for other risk measures.

## B  Further Statements about the Properties

### B.1  Proof of properties of EntRM

*Proof of Lemma 1.*  We only prove the case that $\beta > 0$. The case that $\beta < 0$ follows analogously. For any two independent random variables $X$ and $Y$, we have

$$
\begin{aligned}
U_\beta(X+Y) = \frac{1}{\beta}\log\mathbb{E}[\exp(\beta(X+Y))] &= \frac{1}{\beta}\log\mathbb{E}[\exp(\beta X)\cdot\exp(\beta Y)] \\
&= \frac{1}{\beta}\log\mathbb{E}[\exp(\beta X)] + \frac{1}{\beta}\log\mathbb{E}[\exp(\beta Y)] \\
&= U_\beta(X) + U_\beta(Y),
\end{aligned}
$$

510  therefore ERM is additive.

511  For any two distributions $F_1$ and $F_2$ such that $U_\beta(F_1) > U_\beta(F_2)$, we have

$$U_\beta(F_1) = \frac{1}{\beta} \log \int_{\mathbb{R}} \exp(\beta x) dF_1(x) > \frac{1}{\beta} \log \int_{\mathbb{R}} \exp(\beta x) dF_2(x) = U_\beta(F_1),$$

512  which implies $\int_{\mathbb{R}} \exp(\beta x) dF_1(x) > \int_{\mathbb{R}} \exp(\beta x) dF_2(x)$. Thus for any distribution $G$, it follows that

$$\begin{aligned}
U_\beta(\theta F_1 + (1-\theta)G) &= \frac{1}{\beta} \log \int_{\mathbb{R}} \exp(\beta x) d(\theta F_1(x) + (1-\theta)G(x)) \\
&= \frac{1}{\beta} \log \left( \theta \int_{\mathbb{R}} \exp(\beta x) dF_1(x) + (1-\theta) \int_{\mathbb{R}} \exp(\beta x) dG(x) \right) \\
&> \frac{1}{\beta} \log \left( \theta \int_{\mathbb{R}} \exp(\beta x) dF_2(x) + (1-\theta) \int_{\mathbb{R}} \exp(\beta x) dG(x) \right) \\
&= U_\beta(\theta F_2 + (1-\theta)G).
\end{aligned}$$

513  For any distributions $F$ and $G$ such that $U_\beta(F) > U_\beta(G)$ and $\theta > \theta'$, it holds that

$$\begin{aligned}
&\int_{\mathbb{R}} \exp(\beta x) d(\theta F(x) + (1-\theta)G(x)) - \int_{\mathbb{R}} \exp(\beta x) d(\theta' F(x) + (1-\theta')G(x)) \\
&= (\theta - \theta') \left( \int_{\mathbb{R}} \exp(\beta x) dF(x) - \int_{\mathbb{R}} \exp(\beta x) dG(x) \right) > 0.
\end{aligned}$$

514  Since $t \mapsto \frac{1}{\beta} \log(t)$ is a strictly monotonic mapping, we have $U_\beta(\theta F + (1-\theta)G) > U_\beta(\theta' F + (1-
515  \theta')G)$. Hence ERM satisfies the monotonicity-preserving property. $\qquad\square$

## B.2  Monotonicity preserving

517  We state some lemmas about the monotonicity-preserving property and their proofs here. Note that
518  the results hold for general risk measures satisfying the monotonicity-preserving property. They will
519  be used in the proof of Proposition 1 and Proposition 2.

520  **Lemma 3.** *Let* $\mathrm{T}$ *be a risk measure satisfying the monotonicity-preserving property and* $n \geq 2$
521  *be an arbitrary integer. If* $\mathrm{T}(F_i) \geq \mathrm{T}(G_i), \forall i \in [n]$ *(and* $\mathrm{T}(F_j) \neq \mathrm{T}(G_j)$ *for some* $j \in [n]$*) then*
522  $\mathrm{T}\left(\sum_{i=1}^n \theta_i F_i\right) \geq (>)\mathrm{T}(\sum_{i=1}^n \theta_i G_i)$ *for any* $\theta \in \Delta_n$ *(and* $\theta_j \neq 0$*).*

523  *Proof.* The proof follows from induction. Note that $\sum_{i=1}^n \theta_i F_i = \theta_1 F_1 + (1-\theta_1) \sum_{i=2}^n \frac{\theta_i}{1-\theta_1} F_i$
524  and $\sum_{i=2}^n \frac{\theta_i}{1-\theta_1} F_i \in \mathscr{D}$, therefore by the definition of MP we have $\mathrm{T}(\sum_{i=1}^n \theta_i F_i) \geq \mathrm{T}(\theta_1 G_1 +$
525  $\sum_{i=2}^n \theta_i F_i)$. Suppose that for some $k \in [n-1]$ it holds that $\mathrm{T}(\sum_{i=1}^n \theta_i F_i) \geq \mathrm{T}(\sum_{i=1}^k \theta_i G_i +$
526  $\sum_{i=k+1}^n \theta_i F_i)$. Since

$$\begin{aligned}
\sum_{i=1}^k \theta_i G_i + \sum_{i=k+1}^n \theta_i F_i &= \theta_{k+1} F_{k+1} + \sum_{i=1}^k \theta_i G_i + \sum_{i=k+2}^n \theta_i F_i \\
&= \theta_{k+1} F_{k+1} + (1-\theta_{k+1}) \left[ \sum_{i=1}^k \frac{\theta_i}{1-\theta_{k+1}} G_i + \sum_{i=k+2}^n \frac{\theta_i}{1-\theta_{k+1}} F_i \right]
\end{aligned}$$

and $\frac{1}{1-\theta_{k+1}} \left[ \sum_{i=1}^k \theta_i G_i + \sum_{i=k+2}^n \theta_i F_i \right] \in \mathscr{D}$, it follows that

$$\mathrm{T}\left( \sum_{i=1}^n \theta_i F_i \right) \geq \mathrm{T}\left( \sum_{i=1}^k \theta_i G_i + \sum_{i=k+1}^n \theta_i F_i \right) \geq \mathrm{T}\left( \sum_{i=1}^{k+1} \theta_i G_i + \sum_{i=k+2}^n \theta_i F_i \right).$$

527  The induction is completed. If in addition for some $j \in [n]$ it holds that $\mathrm{T}(F_j) > \mathrm{T}(G_j)$, the proof
528  follows analogously by replacing the inequality to the strict one and the fact that $\theta_j > 0$. $\qquad\square$

**Lemma 4** (Monotonicity-preserving under pairwise transport)**.** *Let* $\mathrm{T}$ *be a risk measure satisfying the monotonicity-preserving property. Suppose $n \geq 2$ and $(F_i)_{i \in [n]}$ satisfies $\mathrm{T}(F_1) \leq \mathrm{T}(F_2)... \leq \mathrm{T}(F_n)$. For any $\theta, \theta' \in \Delta_n$ and any $1 \leq i < j \leq n$ such that*

$$
\begin{cases}
\theta'_i \leq \theta_i, \\
\theta'_j \geq \theta_j, \\
\theta'_k = \theta_k, \quad k \neq i, j
\end{cases}
$$

*It holds that $\mathrm{T}(\sum_{i=1}^n \theta_i F_i) \leq \mathrm{T}(\sum_{i=1}^n \theta'_i F_i)$.*

*Proof.* Observe that

$$
\sum_{k=1}^n \theta'_k F_k = \theta'_i F_i + \theta'_j F_j + \sum_{k \neq i,j} \theta'_k F_k = \theta'_i F_i + \theta'_j F_j + \sum_{k \neq i,j} \theta_k F_k
$$
$$
= (\theta'_i F_i + \theta'_j F_j) + (1 - \theta_i - \theta_j) \sum_{k \neq i,j} \theta_k F_k.
$$

By the definition of the monotonicity-preserving property, it suffices to prove $\mathrm{T}(\frac{1}{\theta_i + \theta_j}(\theta'_i F_i + \theta'_j F_j)) \geq \mathrm{T}(\frac{1}{\theta_i + \theta_j}(\theta_i F_i + \theta_j F_j))$. The result follows from the definition and the fact that $\mathrm{T}(F_i) \leq \mathrm{T}(F_j)$ and $\theta'_i \leq \theta_i$. $\qquad \square$

**Lemma 5** (Monotonicity-preserving under block-wise transport)**.** *Suppose $n \geq 2$ and $(F_i)_{i \in [n]}$ satisfies $\mathrm{T}(F_1) \leq \mathrm{T}(F_2)... \leq \mathrm{T}(F_n)$. It holds that $\mathrm{T}(\sum_{i=1}^n \theta_i F_i) \leq \mathrm{T}(\sum_{i=1}^n \theta'_i F_i)$ for any $\theta, \theta' \in \Delta_n$ satisfying $\exists k \in [n], \theta'_i \leq \theta_i$ if $i \leq k$ and $\theta'_i \geq \theta_i$ otherwise.*

*Proof.* Fix $k \in [n]$. We rewrite the assumption imposed to $\theta'$ as $\theta'_i = \theta_i - \delta_i$ for $i \leq k$ and $\theta'_i = \theta_i + \delta_i$ for $i > k$, where each $\delta_i \geq 0$. It will be shown that there exists a sequence $\{\theta^l\}_{l \in [k]}$ satisfying $\theta^0 = \theta$ and $\theta^k = \theta'$ such that $\mathrm{T}(\theta^l) \leq \mathrm{T}(\theta^{l+1})$, then the proof shall be completed.

The sequence is constructed as follows: at the $l$-th iteration, we transport probability mass $\delta_l$ of $\theta_l$ to the probability mass of $k + 1, ..., n$. Specifically, we start from moving to the least number $i_l \geq i_{l-1}$ that satisfy $\theta_{i_l}^{l-1} < \theta'_{i_l}$ and sequentially move to the next one if there is remaining mass. The iteration stops until all the mass $\delta_l$ are transported. Repeating the procedure for $k$ times we obtain $\theta^k = \theta'$. The inequality $\mathrm{T}(\theta^l) \leq \mathrm{T}(\theta^{l+1})$ for each iteration follows from Lemma 4. $\qquad \square$

### B.3 Proof of properties of EERM

*Proof of Lemma 2.* We only provide the proof for the case $\beta > 0$. The case $\beta < 0$ follows from analogous arguments. For any $F, G \in \mathscr{D}_M$, without loss of generality we assume $\int_0^M G(x) d \exp(\beta x) - \int_0^M F(x) d \exp(\beta x) \geq 0$, otherwise we switch the order.

$$
|E_\beta(F) - E_\beta(G)| = \left| \int_0^M \exp(\beta x) dF(x) - \int_0^M \exp(\beta x) dG(x) \right|
$$
$$
= \left| \exp(\beta x) F(x) |_0^M - \int_0^M F(x) d \exp(\beta x) - \exp(\beta x) G(x) |_0^M + \int_0^M G(x) d \exp(\beta x) \right|
$$
$$
= \int_0^M (G(x) - F(x)) d \exp(\beta x)
$$
$$
\leq \int_0^M |G(x) - F(x)| \, d \exp(\beta x)
$$
$$
\leq \|F - G\|_\infty \int_0^M 1 d \exp(\beta x)
$$
$$
= (\exp(\beta M) - 1) \|F - G\|_\infty.
$$

To show the tightness of the constant, consider two scaled Bernoulli distributions $F = (1 - \mu_1)\psi_0 + \mu_1\psi_M$ and $G = (1 - \mu_2)\psi_0 + \mu_2\psi_M$ with $\Delta := \mu_1 - \mu_2 > 0$, where $\mu_1, \mu_2 \in (0, 1)$ are some constants to be determined. It holds that

$$
\begin{aligned}
E_\beta(F) - E_\beta(G) &= \mu_1 \exp(\beta M) + 1 - \mu_1 - (\mu_2 \exp(\beta M) + 1 - \mu_2) \\
&= (\mu_1 - \mu_2)(\exp(\beta M) - 1) \\
&= \|F - G\|_\infty (\exp(\beta M) - 1).
\end{aligned}
$$

where the last equality holds since $\|F - G\|_\infty = F(0) - G(0) = \mu_1 - \mu_2 = \Delta$ (independent of $M$). More formally, we have

$$
\inf_{M>0, \beta>0} \sup_{F,G \in \mathscr{D}_M} \frac{|E_\beta(F) - E_\beta(G)|}{\|F - G\|_\infty} = \exp(\beta M) - 1.
$$

$\square$

# C  Algorithms for the Random Reward

We present the algorithms for the random reward in this section, which share the same intuitions as the deterministic reward case. Therefore we focus on clarifying their differences here. We denote by $\delta(\cdot)$ the Dirac delta function.

## C.1  RODI-MF

In each episode, the algorithm includes the planning phase (Line 4-12) and the interaction phase (Line 13-17). We highlight two key differences in the planning phase. We introduce the superscript $k$ to the variables of Algorithm 4 in episode $k$. The first difference is that the algorithm *implicitly* maintains the empirical reward distribution in addition to the empirical transition model

$$
\hat{\mathcal{R}}_h^k(s, a) = \frac{\sum_{\tau \in [k-1]} \mathbb{I}_h^\tau(s, a)\delta(\cdot - R_h^\tau)}{N_h^k(s, a)}.
$$

Analogous to the previous setting, we claim that Line 6 is equivalent to a model-based Bellman update for those visited $(s, a)$s. Fix an $(s, a, k, h)$ such that $N_h^k(s, a) \geq 1$. We have shown that for any $\nu \in \mathscr{D}^{\mathcal{S}}$,

$$
\left[\hat{P}_h^k \nu\right](s, a) = \frac{1}{N_h^k(s, a)} \sum_{\tau \in [k-1]} \mathbb{I}_h^\tau(s, a)\nu(s_{h+1}^\tau).
$$

Hence the update formula in Line 6 of Algorithm 4 can be rewritten as

$$
\eta_h^k(s, a) = \left[\hat{P}_h^k \nu_h^k\right](s, a) * \hat{\mathcal{R}}_h^k(s, a) = [\mathcal{B}(\hat{P}_h^k, \hat{\mathcal{R}}_h^k)\nu](s, a).
$$

Alternatively, the unvisited $(s, a)$ remains to be the return distribution corresponding to the highest possible reward $H + 1 - h$. The second difference is that the optimism constant $c_h^k(s, a)$ is increased by an amount of $\sqrt{\frac{1}{2N_h^k(s,a)\vee 1}\iota}$, which corresponds to the estimation error arisen from the unknown reward distribution. The additional term is a lower order term, implying that the regret upper bound of Algorithm 4 is in the same order as that of Algorithm 1.

---

**Algorithm 4** `RODI-MF` (for the random reward)

---

1: Input: $T$ and $\delta$
2: Initialize $N_h(\cdot,\cdot) \leftarrow 0; \eta_h(\cdot,\cdot), \nu_h(\cdot) \leftarrow \psi_{H+1-h}$ for all $h \in [H]$
3: **for** $k = 1 : K$ **do**
4:     **for** $h = H : 1$ **do**
5:         **if** $N_h(\cdot,\cdot) > 0$ **then**
6:             $\eta_h(\cdot,\cdot) \leftarrow \frac{1}{(N_h(\cdot,\cdot))^2} \sum_{\tau,\tau' \in [k-1]^2} \mathbb{I}_h^\tau(\cdot,\cdot) \mathbb{I}_h^{\tau'}(\cdot,\cdot) \nu_{h+1}(s_{h+1}^\tau)(\cdot - R_h^{\tau'}(\cdot,\cdot))$
7:         **end if**
8:         $c_h(\cdot,\cdot) \leftarrow \sqrt{\frac{2S}{N_h(\cdot,\cdot)\vee 1}\iota} + \sqrt{\frac{1}{2N_h(\cdot,\cdot)\vee 1}\iota}$
9:         $\eta_h(\cdot,\cdot) \leftarrow O^\infty_{c_h(\cdot,\cdot)}\eta_h(\cdot,\cdot)$
10:        $\pi_h(\cdot) \leftarrow \arg\max_a U_\beta(\eta_h(\cdot,a))$
11:        $\nu_h(\cdot) \leftarrow \eta_h(\cdot, \pi_h(\cdot))$
12:     **end for**
13:     Receive $s_1^k$
14:     **for** $h = 1 : H$ **do**
15:         $a_h^k \leftarrow \pi_h(s_h^k)$ and transit to $s_{h+1}^k$
16:         $N_h(s_h^k, a_h^k) \leftarrow N_h(s_h^k, a_h^k) + 1$
17:     **end for**
18: **end for**

---

## C.2 RODI-MB

We provide a model-based algorithm (Algorithm 5), which is equivalent to a *nearly classical* algorithm (Algorithm 5). We emphasize the difference between Algorithm 5 and Algorithm 2. For each $(s,a)$, it applies the distributional optimism operators $O^1_{c^k_{h,1}(s,a)}$ and $O^\infty_{c^k_{h,2}(s,a)}$ to the empirical transition model $\hat{P}^k_h(s,a)$ and the empirical reward distribution $\hat{\mathcal{R}}^k_h(s,a)$ respectively, in which $c^k_{h,1}(s,a)$ and $c^k_{h,2}(s,a)$ are set to be $\sqrt{\frac{2S}{N^k_h(s,a)\vee 1}\iota}$ and $\sqrt{\frac{1}{2N^k_h(s,a)\vee 1}\iota}$. Note that the $c^k_{h,2}(s,a)$ is a lower order term in comparison to $c^k_{h,1}(s,a)$, implying that the regret upper bound of Algorithm 5 is in the same order as that of Algorithm 2.

**Remark 1.** *Algorithm 5 is not a fully classical algorithm because it explicitly maintains the reward distributions for all state-action pairs. However, it does not involve the distributional Bellman update that takes the return distributions for all states as input and outputs the return distributions for all state-action pairs. Hence it still reduces considerable computation complexity and space complexity, which makes more close to the classical algorithm rather than the distributional algorithm.*

**Equivalence to** `ROVI`    Define the exponential value functions $W_h(s) \triangleq E_\beta(\nu_h(s))$ and $J_h(s,a) \triangleq E_\beta(\eta_h(s,a))$ for all $(s,a,h)$s. Observe that for two independent r.v.s $X \sim F$ and $Y \sim G$, we have

$$E_\beta(F * g) = E_\beta(X + Y) = E_\beta(X)E_\beta(Y),$$

where $g$ is the PDF of G. Applying EERM to Equation 2 yields the exponential Bellman equation

$$
\begin{aligned}
J_h^*(s,a) &= E_\beta(R_h(s,a))[P_h W_{h+1}^*](s,a), \\
W_h^*(s) &= \text{sign}(\beta) \max_a \text{sign}(\beta) J_h^*(s,a), \ W_{H+1}^*(s) = 1.
\end{aligned}
\tag{5}
$$

We will show that $J_h^k$ in Algorithm 6 corresponds to the exponential value function of $\eta_h^k$ in Algorithm 5. Observe that

$$
\begin{aligned}
E_\beta(\eta_h^k(s,a)) &= E_\beta\left(\left[\tilde{P}_h^k \nu_{h+1}^k\right](s,a) * \tilde{\mathcal{R}}_h^k(s,a)\right) = E_\beta(\tilde{\mathcal{R}}_h^k(s,a)) \cdot \left[\tilde{P}_h^k E_\beta(\nu_{h+1}^k)\right](s,a) \\
&= E_\beta(\tilde{\mathcal{R}}_h^k(s,a))\left[\tilde{P}_h^k W_{h+1}^k\right](s,a) = J_h^k(s,a).
\end{aligned}
$$

The two algorithms generate the policy sequence in the same way. The formal statement is given in Appendix E.

| **Algorithm 5** RODI-MB | **Algorithm 6** ROVI |
|---|---|
| 1: Input: $T$ and $\delta$ | 1: Input: $T$ and $\delta$ |
| 2: $N_h^1(\cdot,\cdot) \leftarrow 0$; $(\hat{P}_h^1(\cdot,\cdot), \hat{\mathcal{R}}_h^1(\cdot,\cdot)) \leftarrow (\frac{1}{S}\mathbf{1}, \psi_{\frac{1}{2}})$ for all $h \in [H]$ | 2: $N_h^1(\cdot,\cdot) \leftarrow 0$; $(\hat{P}_h^1(\cdot,\cdot), \hat{\mathcal{R}}_h^1(\cdot,\cdot)) \leftarrow (\frac{1}{S}\mathbf{1}, \psi_{\frac{1}{2}})$ for all $h \in [H]$ |
| 3: **for** $k = 1 : K$ **do** | 3: **for** $k = 1 : K$ **do** |
| 4: $\quad \nu_{H+1}^k(\cdot) \leftarrow \psi_0$ | 4: $\quad W_{H+1}^k(\cdot) \leftarrow 1$ |
| 5: $\quad$ **for** $h = H : 1$ **do** | 5: $\quad$ **for** $h = H : 1$ **do** |
| 6: $\quad\quad \tilde{P}_h^k(\cdot,\cdot) \leftarrow \mathrm{O}_{c_{h,1}^k(\cdot,\cdot)}^1 \hat{P}_h^k(\cdot,\cdot)$ | 6: $\quad\quad \tilde{P}_h^k(\cdot,\cdot) \leftarrow \mathrm{O}_{c_{h,1}^k(\cdot,\cdot)}^1 \hat{P}_h^k(\cdot,\cdot)$ |
| 7: $\quad\quad \tilde{\mathcal{R}}_h^k(\cdot,\cdot) \leftarrow \mathrm{O}_{c_{h,2}^k(\cdot,\cdot)}^\infty \hat{\mathcal{R}}_h^k(\cdot,\cdot)$ | 7: $\quad\quad \tilde{\mathcal{R}}_h^k(\cdot,\cdot) \leftarrow \mathrm{O}_{c_{h,2}^k(\cdot,\cdot)}^\infty \hat{\mathcal{R}}_h^k(\cdot,\cdot)$ |
| 8: $\quad\quad \eta_h^k(\cdot,\cdot) \leftarrow [\mathcal{B}(\tilde{P}_h^k, \tilde{\mathcal{R}}_h^k)\nu_{h+1}^k](\cdot,\cdot)$ | 8: $\quad\quad J_h^k(\cdot,\cdot) \leftarrow$ $E_\beta\left(\tilde{\mathcal{R}}_h^k(\cdot,\cdot)\right)\left[\tilde{P}_h^k W_{h+1}^k\right](\cdot,\cdot)$ |
| 9: $\quad\quad \pi_h^k(\cdot) \leftarrow \arg\max_a E_\beta(\eta_h^k(\cdot, a))$ | 9: $\quad\quad W_h^k(\cdot) \leftarrow \max_a J_h^k(\cdot, a)$ |
| 10: $\quad\quad \nu_h^k(\cdot) \leftarrow \eta_h^k(\cdot, \pi_h^k(\cdot))$ | 10: $\quad$ **end for** |
| 11: $\quad$ **end for** | 11: $\quad$ Receive $s_1^k$ |
| 12: $\quad$ Receive $s_1^k$ | 12: $\quad$ **for** $h = 1 : H$ **do** |
| 13: $\quad$ **for** $h = 1 : H$ **do** | 13: $\quad\quad a_h^k \leftarrow \arg\max_a J_h^k(s_h^k, a)$ and transit to $s_{h+1}^k$ |
| 14: $\quad\quad a_h^k \leftarrow \pi_h^k(s_h^k)$ and transit to $s_{h+1}^k$ | 14: $\quad\quad$ Compute $N_h^{k+1}(\cdot,\cdot)$ and $\hat{P}_h^{k+1}(\cdot,\cdot)$ |
| 15: $\quad\quad$ Compute $N_h^{k+1}(\cdot,\cdot)$, $\hat{P}_h^{k+1}(\cdot,\cdot)$ and $\hat{\mathcal{R}}_h^{k+1}(\cdot,\cdot)$ | 15: $\quad$ **end for** |
| 16: $\quad$ **end for** | 16: **end for** |
| 17: **end for** | |

# D  Proof of Regret Bounds

## D.1  Proof of Theorem 1

We only prove the case that the reward is random and $\beta > 0$. The proof can be readily adapted to other cases.

**Step 1: Verify optimism.**  Denote by $\iota = \log(2SAT/\delta)$. For any $\delta \in (0,1)$, we define the good event as

$$
\mathcal{G}_\delta := \left\{ \left\| \hat{\mathcal{R}}_h^k(s,a) - \mathcal{R}_h(s,a) \right\|_\infty \leq \sqrt{\frac{1}{2(N_h^k(s,a) \vee 1)}\iota}, \left\| \hat{P}_h^k(\cdot|s,a) - P_h(\cdot|s,a) \right\|_1 \right.
$$
$$
\left. \leq \sqrt{\frac{2S}{N_h^k(s,a) \vee 1}\iota}, \forall (s,a,k,h) \in \mathcal{S} \times \mathcal{A} \times [K] \times [H] \right\},
$$

under which the empirical distributions concentrates around the true distributions w.r.t. $\|\cdot\|_1$.

**Lemma 6** (High probability good event). *For any $\delta \in (0,1)$, the event $\mathcal{G}_\delta$ is true with probability at least $1 - \delta$.*

**Fact 1.** *Let $X$ be a random variable taking values over positive integers and $E$ be an event. If $\mathbb{P}(E|X = i) \geq p$ for any $i = 1, 2, ...$, then $\mathbb{P}(E|X > 0) \geq p$.*

*Proof.*  $\mathbb{P}(E|X > 0) = \frac{\mathbb{P}(E, X>0)}{\mathbb{P}(X>0)} = \frac{\sum_{i\geq 1}\mathbb{P}(E|X=i)\mathbb{P}(X=i)}{\sum_{i\geq 1}\mathbb{P}(X=i)} \geq \frac{\sum_{i\geq 1}p\mathbb{P}(X=i)}{\sum_{i\geq 1}\mathbb{P}(X=i)} = p.$ $\qquad\square$

*Proof.* Fix some $(s,a,k,h) \in \mathcal{S} \times \mathcal{A} \times [K] \times [H]$. If $N_h^k(s,a) = 0$, then we have $(\hat{P}_h^k(\cdot|s,a), \hat{\mathcal{R}}_h^k(s,a)) = (\frac{1}{S}\mathbf{1}, \psi_{\frac{1}{2}})$. A simple calculation yields that for any $\mathcal{R}_h(s,a) \in \mathscr{D}([0,1])$ and any $P_h(\cdot|s,a)$

$$
\left\| \psi_{\frac{1}{2}} - \mathcal{R}_h(s,a) \right\|_\infty \leq \frac{1}{2} \leq \sqrt{\frac{1}{2}\log(2SAT/\delta)}, \left\| \frac{1}{S}\mathbf{1} - P_h(\cdot|s,a) \right\|_1 \leq 2 \leq \sqrt{2S\log(2SAT/\delta)}.
$$

It follows that

$$\mathbb{P}\left(\left\|\hat{\mathcal{R}}_h^k(s,a) - \mathcal{R}_h(s,a)\right\|_\infty \le \sqrt{\frac{1}{2(N_h^k(s,a) \vee 1)} \log(2/\delta)}, \left\|\hat{P}_h^k(\cdot|s,a) - P_h(\cdot|s,a)\right\|_1 \right.$$
$$\left. \le \sqrt{\frac{2S}{N_h^k(s,a) \vee 1} \log(2/\delta)} \,\middle|\, N_h^k(s,a) = 0\right) = 1.$$

Thus the the event is true for the unseen state-action pairs. Now we consider the case that $N_h^k(s,a) > 0$. By the DKW inequality, $\ell_1$ concentration bound of empirical measure and a union bound, we have that for any $n \ge 1$

$$\mathbb{P}\left(\left\|\hat{\mathcal{R}}_h^k(s,a) - \mathcal{R}_h(s,a)\right\|_\infty \le \sqrt{\frac{1}{2N_h^k(s,a)}}, \left\|\hat{P}_h^k(\cdot|s,a) - P_h(\cdot|s,a)\right\|_1 \right.$$
$$\left. \le \sqrt{\frac{2S}{N_h^k(s,a)} \log(2/\delta)} \,\middle|\, N_h^k(s,a) = n\right) \ge 1 - \delta.$$

We use Fact 1 to get

$$\mathbb{P}\left(\left\|\hat{\mathcal{R}}_h^k(s,a) - \mathcal{R}_h(s,a)\right\|_\infty \le \sqrt{\frac{1}{2N_h^k(s,a)} \log(2/\delta)}, \left\|\hat{P}_h^k(\cdot|s,a) - P_h(\cdot|s,a)\right\|_1 \right.$$
$$\left. \le \sqrt{\frac{2S}{N_h^k(s,a)} \log(2/\delta)} \,\middle|\, N_h^k(s,a) > 0\right) \ge 1 - \delta.$$

Taking the two cases into consideration

$$\mathbb{P}\left(\left\|\hat{\mathcal{R}}_h^k(s,a) - \mathcal{R}_h(s,a)\right\|_\infty \le \sqrt{\frac{\log(2/\delta)}{2N_h^k(s,a)}}, \left\|\hat{P}_h^k(\cdot|s,a) - P_h(\cdot|s,a)\right\|_1 \le \sqrt{\frac{2S\log(2/\delta)}{N_h^k(s,a)}}\right)$$
$$= \mathbb{P}\left(\left\|\hat{\mathcal{R}}_h^k(s,a) - \mathcal{R}_h(s,a)\right\|_\infty \le \sqrt{\frac{\log(2/\delta)}{2(N_h^k(s,a) \vee 1)}}, \left\|\hat{P}_h^k(\cdot|s,a) - P_h(\cdot|s,a)\right\|_1 \right.$$
$$\left. \le \sqrt{\frac{2S\log(2/\delta)}{N_h^k(s,a) \vee 1}} \,\middle|\, N_h^k(s,a) = 0\right) \mathbb{P}(N_h^k(s,a) = 0)$$
$$+ \mathbb{P}\left(\left\|\hat{\mathcal{R}}_h^k(s,a) - \mathcal{R}_h(s,a)\right\|_\infty \le \sqrt{\frac{\log(2/\delta)}{2N_h^k(s,a)}}, \left\|\hat{P}_h^k(\cdot|s,a) - P_h(\cdot|s,a)\right\|_1 \le \sqrt{\frac{2S\log(2/\delta)}{N_h^k(s,a)}}\right.$$
$$\left.\middle| N_h^k(s,a) > 0\right)\mathbb{P}(N_h^k(s,a) > 0)$$
$$\ge \mathbb{P}\left(N_h^k(s,a) = 0\right) + (1-\delta)\mathbb{P}(N_h^k(s,a) > 0) \ge 1 - \delta.$$

Applying a union bound over all $(s,a,k,h) \in \mathcal{S} \times \mathcal{A} \times [K] \times [H]$ and rescaling $\delta$ leads to the result. □

Lemma 6 suggests that $\mathcal{G}_\delta$ holds with probability $1 - \delta$, therefore it suffices to prove the theorem conditioned on $\mathcal{G}_\delta$.

**Lemma 7.** *Let* $\mathrm{T}$ *be a functional (not necessarily a risk measure) satisfying the monotonicity, i.e.,* $\mathrm{T}(F) \le \mathrm{T}(G)$ *for any* $F \preceq G$*. For any* $G \in \mathscr{D}([a,b])$*, it holds that if* $G \in B_\infty(F,c)$*, then* $G \preceq \mathrm{O}_c^\infty F$*. Moreover, it holds that*

$$\mathrm{O}_c^\infty F \in \arg\max_{G \in B_\infty(F,c) \cap \mathscr{D}([a,b])} \mathrm{T}(G).$$

*Proof.* Let $G \in \mathscr{D}([a,b]) \cap B_\infty(F,c)$. It follows from the definition of $B_\infty(F,c)$ that $\sup_{x \in [a,b]} |F(x) - G(x)| \le c$, therefore for any $x \in [a,b]$, $G(x) \ge \max(F(x) - c, 0) = (\mathrm{O}_c^\infty F)(x)$. The monotonicity of $\mathrm{T}$ leads to the result. □

627 Notice that $E_\beta$ is also monotonic, which will be used to establish the optimism of the EERM value
628 sequence generated by the algorithm.

629 **Lemma 8.** *For any two distributions $F, G \in \mathscr{D}_M$ and any function $u : \mathbb{R} \to \mathbb{R}$, we have that*

$$|\mathbb{E}_F[u(X)] - \mathbb{E}_G[u(X)]| \leq |u(M) - u(0)| \|F - G\|_\infty.$$

630 *Proof.* Observe that

$$|\mathbb{E}_F[u(X)] - \mathbb{E}_G[u(X)]| = \left| \int_0^M u(x)dF(x) - \int_0^M u(x)dG(x) \right|$$

$$= \left| u(x)F(x)|_0^M - \int_0^M F(x)du(x) - u(x)G(x)|_0^M + \int_0^M G(x)du(x) \right|$$

$$= \left| \int_0^M G(x) - F(x)du(x) \right|$$

$$\leq \left| \int_0^M du(x) \right| \|F - G\|_\infty = |u(M) - u(0)| \|F - G\|_\infty.$$

631 $\qquad\qquad\qquad\qquad\qquad\qquad\qquad\qquad\qquad\qquad\qquad\qquad\qquad\qquad\qquad\qquad\qquad\square$

632 **Lemma 9** (Bound on the optimistic constant)**.** *For any bounded distributions $\{F_i\}_{i \in [n]}$, any $G, G' \in$*
633 *$\mathscr{D}([0,1])$ and any $\theta, \theta' \in \Delta_n$ it holds that if $c \geq \|\theta - \theta'\|_1 + \|G - G'\|_\infty$, then*

$$g * \sum_{i=1}^n \theta_i F_i \preceq \mathrm{O}_c^\infty \left( g' * \sum_{i=1}^n \theta_i' F_i \right),$$

634 *where $g$ and $g'$ are the PDF of $G$ and $G'$ resp..*

635 *Proof.* Without loss of generality assume $F \in \mathscr{D}_M^n$. For any $x \in [0, M + 1)$,

$$\mathrm{O}_c^\infty \left( g' * \sum_{i=1}^n \theta_i' F_i \right)(x) = \left[ \sum_{i=1}^n \theta_i' \int_0^1 F_i(x - r)g'(r)dr - c \right]^+$$

$$= \left[ \sum_{i=1}^n \theta_i \int_0^1 F_i(x - r)g(r)dr + \sum_{i=1}^n \theta_i' \int_0^1 F_i(x - r)g'(r)dr - \sum_i \theta_i \int_0^1 F_i(x - r)g(r)dr - c \right]^+$$

$$= \left[ \left( g * \sum_{i=1}^n \theta_i F_i \right)(x) + \sum_{i=1}^n \theta_i' \int_0^1 F_i(x - r)g'(r)dr - \sum_{i=1}^n \theta_i \int_0^1 F_i(x - r)g(r)dr - c \right]^+.$$

636 It suffices to prove

$$c \geq \left| \sum_{i=1}^n \theta_i' \int_0^1 F_i(x - r)g'(r)dr - \sum_{i=1}^n \theta_i \int_0^1 F_i(x - r)g(r)dr \right|, \forall x \in [0, M + 1].$$

637 We have $\forall x \in [0, M + 1]$,

$$\mathrm{RHS} \leq \left| \sum_{i=1}^n \theta_i' \int_0^1 F_i(x - r)g'(r)dr - \sum_{i=1}^n \theta_i \int_0^1 F_i(x - r)g'(r)dr \right|$$

$$+ \left| \sum_{i=1}^n \theta_i \int_0^1 F_i(x - r)g'(r)dr - \sum_{i=1}^n \theta_i \int_0^1 F_i(x - r)g(r)dr \right|$$

$$\leq \sum_{i=1}^n |\theta_i' - \theta_i| \int_0^1 F_i(x - r)g'(r)dr + \sum_{i=1}^n \theta_i \left| \int_0^1 F_i(x - r)g'(r)dr - \int_0^1 F_i(x - r)g(r)dr \right|$$

$$\leq \|\theta' - \theta\|_1 + \|G - G'\|_\infty,$$

where the last inequality follows from that $\int_0^1 F_i(x-r)g'(r)dr \leq \int_0^1 g'(r)dr = 1$ and the fact that

$$\left| \int_0^1 F_i(x-r)g'(r)dr - \int_0^1 F_i(x-r)g(r)dr \right| = |\mathbb{E}_G[F_i(x-R)] - \mathbb{E}_{G'}[F_i(x-R)]| \leq \|G - G'\|_\infty$$

due to Lemma 8. $\qquad\square$

We define the EERM value produced by the algorithm as $W_h^k(s) \triangleq E_\beta(\nu_h^k(s))$ and $J_h^k(s,a) \triangleq E_\beta(\eta_h^k(s,a))$ for all $(s,a,k,h)$s. Similarly, we define $W_h^*(s) \triangleq E_\beta(\nu_h^*(s))$ and $J_h^*(s,a) \triangleq E_\beta(\eta_h^*(s,a))$ for all $(s,a,h)$s. Using Lemma 9, the monotonicity of EERM, and inductions, we arrives at Lemma 10, which guarantees the sequence $\{W_1^k(s_1^k)\}_{k\in[K]}$ produced by Algorithm 4 is indeed optimistic compared to the optimal value $\{W_1^*(s_1^k)\}_{k\in[K]}$.

**Lemma 10** (Optimism). *Conditioned on event $\mathcal{G}_\delta$, the sequence $\{W_1^k(s_1^k)\}_{k\in[K]}$ produced by Algorithm 4 are all greater than or equal to $W_1^*(s_1^k)$, i.e.,*

$$W_1^k(s_1^k) = E_\beta(\nu_1^k(s_1^k)) \geq E_\beta(\nu_1^*(s_1^k)) = W_1^*(s_1^k), \forall k \in [K].$$

*Proof.* The proof follows from induction. Fix $k \in [K]$. For $h = H$ we have that for any $(s,a)$

$$
\begin{aligned}
J_H^k(s,a) = E_\beta(\eta_H^k(s,a)) &= E_\beta(\mathrm{O}_{c_H^k(s,a)}^\infty(\hat{\mathcal{R}}_H^k(s,a))) \\
&\geq E_\beta(\mathcal{R}_H(s,a)) = J_H^*(s,a),
\end{aligned}
$$

where the inequality is due to Lemma 7 and the fact that $\mathcal{R}_H(s,a) \in B_\infty(\hat{\mathcal{R}}_H(s,a), c_H^k(s,a)) \cap \mathscr{D}_1$. Thus $W_H^k(s) = \max_a J_H^k(s,a) \geq \max_a J_H^*(s,a) = W_H^*(s), \forall s$. Now suppose for $h + 1 \in [2 : H]$, it holds that $W_{h+1}^k(s) \geq W_{h+1}^*(s), \forall s$. For each $(s,a)$, we applying Lemma **??** with $\theta = P_h(s,a), \theta' = \hat{P}_h^k(s,a), F = \nu_{h+1}^k, G = \mathcal{R}_h(s,a)$ and $G' = \hat{\mathcal{R}}_h^k(s,a)$ to obtain

$$[P_h \nu_{h+1}^k](s,a) * f_{\mathcal{R}_h(s,a)} \preceq \mathrm{O}_{c_h^k(s,a)}^\infty([\hat{P}_h^k \nu_{h+1}^k](s,a) * f_{\hat{\mathcal{R}}_h^k(s,a)})$$

since $c_h^k(s,a) = \sqrt{\frac{2S}{N_h^k(s,a)\vee 1}\iota} + \sqrt{\frac{1}{2(N_h^k(s,a)\vee 1)}\iota} \geq \left\| P_h(\cdot|s,a) - \hat{P}_h^k(\cdot|s,a) \right\|_1 + \left\| \mathcal{R}_h(s,a) - \hat{\mathcal{R}}_h^k(s,a) \right\|_\infty$ for $h \in [H-1]$. It follows that

$$
\begin{aligned}
J_h^k(s,a) &= E_\beta(\mathrm{O}_{c_h^k(s,a)}^\infty([\hat{P}_h^k \nu_{h+1}^k](s,a) * f_{\hat{\mathcal{R}}_h^k(s,a)})) \\
&\geq E_\beta([P_h \nu_{h+1}^k](s,a) * f_{\mathcal{R}_h(s,a)}) \\
&= E_\beta(\mathcal{R}_h(s,a)) \cdot [P_h W_{h+1}^k](s,a) \\
&\geq E_\beta(\mathcal{R}_h(s,a)) \cdot [P_h W_{h+1}^*](s,a) \\
&= J_h^*(s,a), \forall(s,a),
\end{aligned}
$$

where the first inequality is due to the property (**M**), and the second inequality follows from the induction assumption. The second equality is due to Equation **??**. Finally it follows that for any $s$,

$$W_h^k(s) = \max_a J_h^k(s,a) \geq \max_a J_h^*(s,a) = W_h^*(s).$$

The induction is completed. $\qquad\square$

**Step 2: Regret decomposition.**

**Lemma 11.** *For any $F_i \in \mathscr{D}$ and any $\theta, \theta' \in \Delta_n$ with any $n \geq 2$, it holds that*

$$\left\| \sum_{i=1}^n \theta_i F_i - \sum_{i=1}^n \theta'_i F_i \right\|_\infty \leq \|\theta - \theta'\|_1.$$

*Proof.*

$$\left\| \sum_{i=1}^{n} \theta_i F_i - \sum_{i=1}^{n} \theta_i' F_i \right\|_{\infty} = \sup_{x \in \mathbb{R}} \left| \sum_{i=1}^{n} (\theta_i F - \theta_i') F_i(x) \right|$$

$$\le \sup_{x \in \mathbb{R}} \sum_{i=1}^{n} |\theta_i - \theta_i'| F_i(x)$$

$$\le \sum_{i=1}^{n} |\theta_i - \theta_i'|$$

$$= \|\theta - \theta'\|_1 .$$

658 $\qquad\qquad\qquad\qquad\qquad\qquad\qquad\qquad\qquad\qquad\qquad\qquad\qquad\qquad\qquad\square$

659 We define $\Delta_h^k \triangleq W_h^k - W_h^{\pi^k} = E_\beta(\nu_h^k) - E_\beta\left(\nu_h^{\pi^k}\right) \in D_h^S$ with

$$D_h \triangleq [1 - \exp(\beta(H+1-h)), \exp(\beta(H+1-h)) - 1]$$

660 and $\delta_h^k \triangleq \Delta_h^k(s_h^k)$. For any $(s,h)$ and any $\pi$, we let $P_h^\pi(\cdot|s) := P_h(\cdot|s, \pi_h(s))$. Observe that the
661 regret can be bounded as

$$\text{Regret}(K) = \sum_{k=1}^{K} \frac{1}{\beta} \log\left(W_1^*(s_1^k)\right) - \frac{1}{\beta} \log\left(W_1^{\pi^k}(s_1^k)\right)$$

$$= \sum_{k=1}^{K} \frac{1}{\beta} \log\left(W_1^*(s_1^k)\right) - \frac{1}{\beta} \log\left(V_1^k(s_1^k)\right) + \frac{1}{\beta} \log\left(W_1^k(s_1^k)\right) - \frac{1}{\beta} \log\left(V_1^{\pi^k}(s_1^k)\right)$$

$$\le \sum_{k=1}^{K} \frac{1}{\beta} \log\left(W_1^k(s_1^k)\right) - \frac{1}{\beta} \log\left(W_1^{\pi^k}(s_1^k)\right)$$

$$\le \frac{1}{\beta} \sum_{k=1}^{K} W_1^k(s_1^k) - W_1^{\pi^k}(s_1^k) = \frac{1}{\beta} \sum_{k=1}^{K} \delta_1^k.$$

662 We can decompose $\delta_h^k$ as follows

$$\delta_h^k = E_\beta\left(\nu_h^k(s_h^k)\right) - E_\beta\left(\nu_h^{\pi^k}(s_h^k)\right)$$

$$= E_\beta\left(O_{c_h^k}\left(\left[\hat{P}_h^{\pi^k} \eta_{h+1}^k\right](s_h^k) * f_{\hat{\mathcal{R}}_h^{\pi^k}(s_h^k)}\right)\right) - E_\beta\left(\left[P_h^{\pi^k} \nu_{h+1}^{\pi^k}\right](s_h^k) * f_{\mathcal{R}_h^{\pi^k}(s_h^k)}\right)$$

$$= \underbrace{E_\beta\left(O_{c_h^k}\left(\left[\hat{P}_h^{\pi^k} \nu_{h+1}^k\right](s_h^k) * f_{\hat{\mathcal{R}}_h^{\pi^k}(s_h^k)}\right)\right) - E_\beta\left(\left[\hat{P}_h^{\pi^k} \nu_{h+1}^k\right](s_h^k) * f_{\hat{\mathcal{R}}_h^{\pi^k}(s_h^k)}\right)}_{(a)}$$

$$+ \underbrace{E_\beta\left(\left[\hat{P}_h^{\pi^k} \nu_{h+1}^k\right](s_h^k) * f_{\hat{\mathcal{R}}_h^{\pi^k}(s_h^k)}\right) - E_\beta\left(\left[\hat{P}_h^{\pi^k} \nu_{h+1}^k\right](s_h^k) * f_{\mathcal{R}_h^{\pi^k}(s_h^k)}\right)}_{(b)}$$

$$+ \underbrace{E_\beta\left(\left[\hat{P}_h^{\pi^k} \nu_{h+1}^k\right](s_h^k) * f_{\mathcal{R}_h^{\pi^k}(s_h^k)}\right) - E_\beta\left(\left[P_h^{\pi^k} \nu_{h+1}^k\right](s_h^k) * f_{\mathcal{R}_h^{\pi^k}(s_h^k)}\right)}_{(c)}$$

$$+ \underbrace{E_\beta\left(\left[P_h^{\pi^k} \nu_{h+1}^k\right](s_h^k) * f_{\mathcal{R}_h^{\pi^k}(s_h^k)}\right) - E_\beta\left(\left[P_h^{\pi^k} \nu_{h+1}^{\pi^k}\right](s_h^k) * f_{\mathcal{R}_h^{\pi^k}(s_h^k)}\right)}_{(d)} .$$

663 Using the Lipschitz property of EERM,

$$(a) \le L_{H+1-h} \left\| O_{c_h^k}^\infty\left(\left[\hat{P}_h^{\pi^k} \nu_{h+1}^k\right](s_h^k) * f_{\hat{\mathcal{R}}_h^{\pi^k}(s_h^k)}\right) - \left[\hat{P}_h^{\pi^k} \nu_{h+1}^k\right](s_h^k) * f_{\hat{\mathcal{R}}_h^{\pi^k}(s_h^k)} \right\|_\infty$$

$$\le L_{H+1-h} c_h^k$$

$$= (\exp(\beta(H+1-h)) - 1)\left(\sqrt{\frac{1}{2(N_h^k \vee 1)}\iota} + \sqrt{\frac{S}{(N_h^k \vee 1)}\iota}\right) .$$

Define $e_h^k \triangleq \left\| \hat{P}_h^k(s_h^k) - P_h^{\pi^k}(s_h^k) \right\|_1$. We can bound $(b)$ as

$$
\begin{aligned}
(b) &= \left( E_\beta(\hat{\mathcal{R}}_h^{\pi^k}(s_h^k)) - E_\beta(\mathcal{R}_h^{\pi^k}(s_h^k)) \right) \cdot E_\beta\left( \left[ \hat{P}_h^k \nu_{h+1}^k \right](s_h^k) \right) \\
&\leq L_1 \left\| \hat{\mathcal{R}}_h^{\pi^k}(s_h^k) - \mathcal{R}_h^{\pi^k}(s_h^k) \right\|_\infty \left[ \hat{P}_h^k W_{h+1}^k \right](s_h^k) \\
&\leq (\exp(\beta) - 1)\sqrt{\frac{1}{2(N_h^k \vee 1)}}\, \iota \exp(\beta(H-h)) \\
&\leq (\exp(\beta(H+1-h)) - 1)\sqrt{\frac{1}{2(N_h^k \vee 1)}}\, \iota.
\end{aligned}
$$

We bound $(c)$ as

$$
\begin{aligned}
(c) &= E_\beta(\mathcal{R}_h^{\pi^k}(s_h^k))\left( E_\beta\left( \left[ \hat{P}_h^{\pi^k} \nu_{h+1}^k \right](s_h^k) \right) - E_\beta\left( \left[ P_h^{\pi^k} \nu_{h+1}^k \right](s_h^k) \right) \right) \\
&\leq \exp(\beta)L_{H-h}\left\| \left[ \hat{P}_h^{\pi^k} \nu_{h+1}^k \right](s_h^k) - \left[ P_h^{\pi^k} \nu_{h+1}^k \right](s_h^k) \right\|_\infty \\
&\leq \exp(\beta)(\exp(\beta(H-h)) - 1)\left\| \hat{P}_h^{\pi^k}(s_h^k) - P_h^{\pi^k}(s_h^k) \right\|_1 \\
&\leq (\exp(\beta(H+1-h)) - 1)\sqrt{\frac{S}{(N_h^k \vee 1)}}\, \iota,
\end{aligned}
$$

where the second inequality is due to Lemma 11. By the linearity of EERM, We bound $(d)$ as

$$
\begin{aligned}
(d) &= E_\beta(\mathcal{R}_h^{\pi^k}(s_h^k))\left[ P_h^{\pi^k}(V_{h+1}^k - V_{h+1}^{\pi^k}) \right](s_h^k) \\
&= E_\beta(\mathcal{R}_h^{\pi^k}(s_h^k))\left[ P_h^{\pi^k} \Delta_{h+1}^k \right](s_h^k) \\
&= E_\beta(\mathcal{R}_h^{\pi^k}(s_h^k))(\epsilon_h^k + \delta_{h+1}^k),
\end{aligned}
$$

where $\epsilon_h^k \triangleq [P_h^{\pi^k} \Delta_{h+1}^k](s_h^k) - \Delta_{h+1}^k(s_{h+1}^k)$ is a martingale difference sequence with $\epsilon_h^k \in 2D_{h+1}$ a.s. for all $(k,h) \in [K] \times [H]$. Since

$$
(b) + (c) \leq L_{H+1-h}c_h^k,
$$

we can bound $\delta_h^k$ recursively as

$$
\delta_h^k \leq 2L_{H+1-h}c_h^k + E_\beta(\mathcal{R}_h^{\pi^k}(s_h^k))(\epsilon_h^k + \delta_{h+1}^k).
$$

Repeating the procedure, we can get

$$
\begin{aligned}
\delta_1^k &\leq 2\sum_{h=1}^{H-1} L_{H+1-h} \prod_{i=1}^{h-1} E_\beta(\mathcal{R}_i^{\pi^k}(s_i^k))c_h^k + \sum_{h=1}^{H-1}\prod_{i=1}^{h} E_\beta(\mathcal{R}_i^{\pi^k}(s_i^k))\epsilon_h^k + \prod_{i=1}^{H-1} E_\beta(\mathcal{R}_i^{\pi^k}(s_i^k))\delta_H^k \\
&\leq 2\sum_{h=1}^{H-1} (\exp(\beta(H+1-h)) - 1)\exp(\beta(h-1))c_h^k + \sum_{h=1}^{H-1}\prod_{i=1}^{h} E_\beta(\mathcal{R}_i^{\pi^k}(s_i^k))\epsilon_h^k + \prod_{i=1}^{H-1} E_\beta(\mathcal{R}_i^{\pi^k}(s_i^k))\delta_H^k \\
&\leq 2\sum_{h=1}^{H-1} (\exp(\beta H) - 1)c_h^k + \sum_{h=1}^{H-1}\prod_{i=1}^{h} E_\beta(\mathcal{R}_i^{\pi^k}(s_i^k))\epsilon_h^k + \prod_{i=1}^{H-1} E_\beta(\mathcal{R}_i^{\pi^k}(s_i^k))\delta_H^k.
\end{aligned}
$$

It follows that

$$
\sum_{k=1}^{K} \delta_1^k \leq 2(\exp(\beta(H+1)) - 1)\sum_{k=1}^{K}\sum_{h=1}^{H-1} c_h^k + \sum_{k=1}^{K}\sum_{h=1}^{H-1}\prod_{i=1}^{h} E_\beta(\mathcal{R}_i^{\pi^k}(s_i^k))\epsilon_h^k + \sum_{k=1}^{K}\prod_{i=1}^{H-1} E_\beta(\mathcal{R}_i^{\pi^k}(s_i^k))\delta_H^k.
$$

 **Step 3: Bound each term.** The first term can be bounded as

$$
2(\exp(\beta(H+1))-1)\sum_{k=1}^{K}\sum_{h=1}^{H-1}c_h^k = 2(\exp(\beta(H+1))-1)\sum_{h=1}^{H-1}\sum_{k=1}^{K}\left(\sqrt{\frac{1}{2(N_h^k\vee 1)}\iota}+\sqrt{\frac{S}{(N_h^k\vee 1)}\iota}\right)
$$

$$
\leq 3(\exp(\beta(H+1))-1)\sum_{h=1}^{H-1}\sqrt{2S^2AK\iota}
$$

$$
= 3(\exp(\beta(H+1))-1)\sqrt{2S^2AK\iota}.
$$

 Observe that

$$
\prod_{i=1}^{h}E_\beta(\mathcal{R}_i^{\pi^k}(s_i^k))\epsilon_h^k \in \exp(\beta h)D_h = \exp(\beta h)[1-\exp(\beta(H+1-h)),\exp(\beta(H+1-h))-1]
$$

$$
\subseteq [1-\exp(\beta(H+1)),\exp(\beta(H+1))-1],
$$

 thus we can bound the second term by Azuma-Hoeffding inequality: with probability at least $1-\delta'$,
 the following holds

$$
\sum_{k=1}^{K}\sum_{h=1}^{H-1}\prod_{i=1}^{h}E_\beta(\mathcal{R}_i^{\pi^k}(s_i^k))\epsilon_h^k \leq (\exp(\beta(H+1))-1)\sqrt{2KH\log(1/\delta')}.
$$

 We have

$$
\sum_{k=1}^{K}\prod_{i=1}^{H-1}E_\beta(\mathcal{R}_i^{\pi^k}(s_i^k))\delta_H^k \leq \sum_{k=1}^{K}\exp(\beta(H-1))L_1 c_H^k
$$

$$
\leq \sum_{k=1}^{K}\exp(\beta(H-1))(\exp(\beta)-1)\left(\sqrt{\frac{1}{2(N_h^k\vee 1)}\iota}+\sqrt{\frac{S}{(N_h^k\vee 1)}\iota}\right)
$$

$$
\leq 1.5(\exp(\beta H)-1)\sqrt{2S^2AK\iota}.
$$

 Using a union bound and let $\delta=\delta'=\frac{\tilde{\delta}}{2}$, we have that with probability at least $1-\delta$,

$$
\text{Regret}(K) \leq \frac{1}{\beta}\left(4.5(\exp(\beta(H+1))-1)\sqrt{2S^2AK\iota}+(\exp(\beta(H+1))-1)\sqrt{2KH\iota}\right)
$$

$$
= \tilde{\mathcal{O}}\left(\frac{\exp(\beta H)-1}{\beta H}H\sqrt{HS^2AT}\right),
$$

 where $\iota \triangleq \log(4SAT/\delta)$.

 ## D.2   Proof of Theorem 2

 We only prove the case that the reward is random. The proof can be readily adapted to the deterministic
 reward case.

 **Distributional analysis vs non-distributional analysis**   By Proposition 5, Algorithm 5 is equivalent
 to Algorithm 6. Since Algorithm 6 is a classical algorithm, it is thus natural to use the classical
 analysis to derive the regret bounds. That being said, we will show that the distributional analysis
 yields a tighter bound than the non-distributional analysis. In particular, **the latter one yields a regret**
 **bound that explodes as $\beta$ approaches zero, but our analysis can recover the desired order when**
 **reduced to the risk-neutral setting.**

 **Step 1: Verify optimism.**   Lemma 6 suggests that $\mathcal{G}_\delta$ holds with probability $1-\delta$, therefore it
 suffices to prove the theorem conditioned on $\mathcal{G}_\delta$.

 **Lemma 12** (Optimistic transition model). *Fix $(s,a,k,h)$. For any $P \in B_1(\hat{P}_h^k(s,a),c_{h,1}^k(s,a))$, we*
 *have*

$$
E_\beta\left(\left[\tilde{P}_h^k\nu_{h+1}^k\right](s,a)\right) \geq E_\beta\left(\left[P\nu_{h+1}^k\right](s,a)\right).
$$

**Lemma 13** (Optimism). *Conditioned on event $\mathcal{G}_\delta$, the sequence $\{W_1^k(s_1^k)\}_{k \in [K]}$ produced by Algorithm 5 are all greater than or equal to $W_1^*(s_1^k)$, i.e.,*

$$W_1^k(s_1^k) = E_\beta(\nu_1^k(s_1^k)) \geq E_\beta(\nu_1^*(s_1^k)) = W_1^*(s_1^k), \forall k \in [K].$$

*Proof.* The proof follows from induction. Fix $k \in [K]$. For $h = H$, we have that for any $(s, a)$

$$J_H^k(s, a) = E_\beta(\eta_H^k(s, a)) = E_\beta(\mathrm{O}_{c_{H,2}^k(s,a)}^\infty(\hat{\mathcal{R}}_H^k(s, a)))$$
$$\geq E_\beta(\mathcal{R}_H(s, a)) = J_H^*(s, a),$$

where the inequality is due to Lemma 7 and the fact that $\mathcal{R}_H(s, a) \in B_\infty(\hat{\mathcal{R}}_H(s, a), c_{H,2}^k(s, a)) \cap \mathscr{D}_1$. Thus $W_H^k(s) = \max_a J_H^k(s, a) \geq \max_a J_H^*(s, a) = W_H^*(s), \forall s$. Now suppose for $h + 1 \in [2 : H]$, it holds that $W_{h+1}^k(s) \geq W_{h+1}^*(s), \forall s$. It follows that

$$J_h^k(s, a) = E_\beta\left(\tilde{\mathcal{R}}_h^k(s, a)\right) E_\beta\left(\left[\tilde{P}_h^k \nu_{h+1}^k\right](s, a)\right)$$
$$\geq E_\beta\left(\mathcal{R}_h(s, a)\right) E_\beta\left(\left[P_h \nu_{h+1}^k\right](s, a)\right)$$
$$\geq E_\beta\left(\mathcal{R}_h(s, a)\right) E_\beta\left(\left[P_h \nu_{h+1}^*\right](s, a)\right)$$
$$= J_h^*(s, a), \forall(s, a),$$

where the first inequality is due to Lemma 12, and the second inequality follows from the induction assumption. Since for any $s$,

$$W_h^k(s) = \max_a J_h^k(s, a) \geq \max_a J_h^*(s, a) = W_h^*(s),$$

The induction is completed. $\qquad\square$

**Step 2: Regret decomposition.** We define $\Delta_h^k \triangleq W_h^k - W_h^{\pi^k} = E_\beta(\nu_h^k) - E_\beta\left(\nu_h^{\pi^k}\right) \in D_h^S$ with

$$D_h \triangleq [1 - \exp(\beta(H + 1 - h)), \exp(\beta(H + 1 - h)) - 1]$$

and $\delta_h^k \triangleq \Delta_h^k(s_h^k)$. For any $(s, h)$ and any $\pi$, we let $P_h^\pi(\cdot|s) \triangleq P_h(\cdot|s, \pi_h(s))$. The regret can be bounded as

$$\mathrm{Regret}(K) = \sum_{k=1}^K \frac{1}{\beta} \log\left(W_1^*(s_1^k)\right) - \frac{1}{\beta} \log\left(W_1^{\pi^k}(s_1^k)\right)$$
$$= \sum_{k=1}^K \frac{1}{\beta} \log\left(W_1^*(s_1^k)\right) - \frac{1}{\beta} \log\left(W_1^k(s_1^k)\right) + \frac{1}{\beta} \log\left(W_1^k(s_1^k)\right) - \frac{1}{\beta} \log\left(W_1^{\pi^k}(s_1^k)\right)$$
$$\leq \sum_{k=1}^K \frac{1}{\beta} \log\left(W_1^k(s_1^k)\right) - \frac{1}{\beta} \log\left(W_1^{\pi^k}(s_1^k)\right)$$
$$\leq \frac{1}{\beta} \sum_{k=1}^K W_1^k(s_1^k) - W_1^{\pi^k}(s_1^k) = \frac{1}{\beta} \sum_{k=1}^K \delta_1^k.$$

704 We can decompose $\delta_h^k$ as follows

$$
\begin{aligned}
\delta_h^k &= E_\beta(\nu_h^k(s_h^k)) - E_\beta(\nu_h^{\pi^k}(s_h^k)) \\
&= E_\beta\left(\tilde{\mathcal{R}}_h^{\pi^k}(s_h^k)\right) E_\beta\left(\left[\tilde{P}_h^k \nu_{h+1}^k\right](s_h^k)\right) - E_\beta\left(\mathcal{R}_h^{\pi^k}(s_h^k)\right) E_\beta\left(\left[P_h^{\pi^k} \nu_{h+1}^k\right](s_h^k)\right) \\
&= \underbrace{E_\beta\left(\tilde{\mathcal{R}}_h^{\pi^k}(s_h^k)\right) E_\beta\left(\left[\tilde{P}_h^k \nu_{h+1}^k\right](s_h^k)\right) - E_\beta\left(\mathcal{R}_h^{\pi^k}(s_h^k)\right) E_\beta\left(\left[\tilde{P}_h^k \nu_{h+1}^k\right](s_h^k)\right)}_{(a)} \\
&\quad + \underbrace{E_\beta\left(\mathcal{R}_h^{\pi^k}(s_h^k)\right) E_\beta\left(\left[\tilde{P}_h^k \nu_{h+1}^k\right](s_h^k)\right) - E_\beta\left(\mathcal{R}_h^{\pi^k}(s_h^k)\right) E_\beta\left(\left[P_h^{\pi^k} \nu_{h+1}^k\right](s_h^k)\right)}_{(b)} \\
&\quad + \underbrace{E_\beta\left(\mathcal{R}_h^{\pi^k}(s_h^k)\right) E_\beta\left(\left[P_h^{\pi^k} \nu_{h+1}^k\right](s_h^k)\right) - E_\beta\left(\mathcal{R}_h^{\pi^k}(s_h^k)\right) E_\beta\left(\left[P_h^{\pi^k} \nu_{h+1}^{\pi^k}\right](s_h^k)\right)}_{(c)} \\
&= \underbrace{E_\beta\left(\tilde{\mathcal{R}}_h^{\pi^k}(s_h^k)\right)\left[\tilde{P}_h^k W_{h+1}^k\right](s_h^k) - E_\beta\left(\mathcal{R}_h^{\pi^k}(s_h^k)\right)\left[\tilde{P}_h^k W_{h+1}^k\right](s_h^k)}_{(a)} \\
&\quad + \underbrace{E_\beta\left(\mathcal{R}_h^{\pi^k}(s_h^k)\right)\left[\tilde{P}_h^k W_{h+1}^k\right](s_h^k) - E_\beta\left(\mathcal{R}_h^{\pi^k}(s_h^k)\right)\left[P_h^{\pi^k} W_{h+1}^k\right](s_h^k)}_{(b)} \\
&\quad + \underbrace{E_\beta\left(\mathcal{R}_h^{\pi^k}(s_h^k)\right)\left[P_h^{\pi^k} W_{h+1}^k\right](s_h^k) - E_\beta\left(\mathcal{R}_h^{\pi^k}(s_h^k)\right)\left[P_h^{\pi^k} W_{h+1}^{\pi^k}\right](s_h^k)}_{(c)}.
\end{aligned}
$$

705 Both distributional analysis and non-distributional analysis seem to be viable to deal with $(b)$, but the
706 non-distributional analysis turns out to yield an unsatisfactory bound.
707 Non-distributional analysis: Notice that $W_{h+1}^k(s) \le \exp(\beta(H - h))$, $\forall s$. Thus the following holds

$$
\begin{aligned}
(b) &= E_\beta\left(\mathcal{R}_h^{\pi^k}(s_h^k)\right)\left(\left[\tilde{P}_h^k W_{h+1}^k\right](s_h^k) - \left[P_h^{\pi^k} W_{h+1}^k\right](s_h^k)\right) \\
&= E_\beta\left(\mathcal{R}_h^{\pi^k}(s_h^k)\right)\left(\left[\left(\tilde{P}_h^k - P_h^{\pi^k}\right) W_{h+1}^k\right](s_h^k)\right) \\
&\le \exp(\beta)\left\|\tilde{P}_h^k - P_h^{\pi^k}\right\|_1 \max_s W_{h+1}^k(s) \\
&\le 2\exp(\beta(H + 1 - h)) c_{h,1}^k.
\end{aligned}
$$

708 Distributional analysis: Using the Lipschitz property of EERM, we have

$$
\begin{aligned}
(b) &\le L_{H+1-h}\left\|\left[\tilde{P}_h^k \nu_{h+1}^k\right](s_h^k)(\cdot - r_h^k) - \left[P_h^{\pi^k} \nu_{h+1}^k\right](s_h^k)(\cdot - r_h^k)\right\|_\infty \\
&\le L_{H+1-h}\left\|\tilde{P}_h^k - P_h^{\pi^k}\right\|_1 \\
&\le 2 L_{H+1-h} c_{h,1}^k \\
&= 2(\exp(\beta(H + 1 - h)) - 1) c_{h,1}^k,
\end{aligned}
$$

709 where the second inequality is due to Lemma 11. The two types of analysis lead to different
710 coefficients. Consider the risk-neutral setting $\beta \to 0$. For the distributional analysis, the coefficient
711 appears in the regret bound as

$$
\lim_{\beta \to 0} \frac{\exp(\beta(H + 1 - h)) - 1}{\beta} = H + 1 - h,
$$

712 in contrast, the non-distributional analysis leads to that

$$
\lim_{\beta \to 0} \frac{\exp(\beta(H + 1 - h))}{\beta} = \infty.
$$

713 For small $\beta$, the distributional analysis recovers the order of the corresponding risk-neutral algorithm.
714 However, the non-distributional analysis yields a exploding factor as $\beta \to 0$. Therefore, it is not

proper to use the classical analysis to obtain the regret bound of Algorithm 6. We can bound $(a)$ as

$$(a) = \left( E_\beta \left( \tilde{\mathcal{R}}_h^{\pi^k}(s_h^k) \right) - E_\beta \left( \mathcal{R}_h^{\pi^k}(s_h^k) \right) \right) \left[ \tilde{P}_h^k W_{h+1}^k \right](s_h^k)$$

$$\leq L_1 \left\| \tilde{\mathcal{R}}_h^{\pi^k}(s_h^k) - \mathcal{R}_h^{\pi^k}(s_h^k) \right\|_\infty \cdot \exp(\beta(H-h))$$

$$\leq (\exp(\beta(H+1-h)) - 1)c_{h,2}^k,$$

where the second inequality follows from the DKW inequality and the definition of $c_{h,2}^k$. Term $(c)$ is bounded as

$$(c) = E_\beta \left( \mathcal{R}_h^{\pi^k}(s_h^k) \right) \left[ P_h^{\pi^k} (W_{h+1}^k - W_{h+1}^{\pi^k}) \right](s_h^k)$$

$$= E_\beta \left( \mathcal{R}_h^{\pi^k}(s_h^k) \right) \left[ P_h^{\pi^k} \Delta_{h+1}^k \right](s_h^k)$$

$$= E_\beta \left( \mathcal{R}_h^{\pi^k}(s_h^k) \right) (\epsilon_h^k + \delta_{h+1}^k),$$

where $\epsilon_h^k \triangleq [P_h^{\pi^k} \Delta_{h+1}^k](s_h^k) - \Delta_{h+1}^k(s_{h+1}^k)$ is a martingale difference sequence with $\epsilon_h^k \in 2D_{h+1}$ a.s. for all $(k,h) \in [K] \times [H]$. Denote by $c_h^k \triangleq c_{h,1}^k + c_{h,2}^k$. In summary, we can bound $\delta_h^k$ recursively as

$$\delta_h^k \leq 2L_{H+1-h}c_h^k + E_\beta(\mathcal{R}_h^{\pi^k}(s_h^k))(\epsilon_h^k + \delta_{h+1}^k).$$

Repeating the procedure, we can get

$$\delta_1^k \leq 2 \sum_{h=1}^{H-1} L_{H+1-h} \prod_{i=1}^{h-1} E_\beta(\mathcal{R}_i^{\pi^k}(s_i^k))c_h^k + \sum_{h=1}^{H-1} \prod_{i=1}^{h} E_\beta(\mathcal{R}_i^{\pi^k}(s_i^k))\epsilon_h^k + \prod_{i=1}^{H-1} E_\beta(\mathcal{R}_i^{\pi^k}(s_i^k))\delta_H^k$$

$$\leq 2 \sum_{h=1}^{H-1} (\exp(\beta(H+1-h)) - 1) \exp(\beta(h-1))c_h^k + \sum_{h=1}^{H-1} \prod_{i=1}^{h} E_\beta(\mathcal{R}_i^{\pi^k}(s_i^k))\epsilon_h^k + \prod_{i=1}^{H-1} E_\beta(\mathcal{R}_i^{\pi^k}(s_i^k))\delta_H^k$$

$$\leq 2 \sum_{h=1}^{H-1} (\exp(\beta H) - 1)c_h^k + \sum_{h=1}^{H-1} \prod_{i=1}^{h} E_\beta(\mathcal{R}_i^{\pi^k}(s_i^k))\epsilon_h^k + \prod_{i=1}^{H-1} E_\beta(\mathcal{R}_i^{\pi^k}(s_i^k))\delta_H^k.$$

It follows that

$$\sum_{k=1}^{K} \delta_1^k \leq 2(\exp(\beta(H+1)) - 1) \sum_{k=1}^{K} \sum_{h=1}^{H-1} c_h^k + \sum_{k=1}^{K} \sum_{h=1}^{H-1} \prod_{i=1}^{h} E_\beta(\mathcal{R}_i^{\pi^k}(s_i^k))\epsilon_h^k + \sum_{k=1}^{K} \prod_{i=1}^{H-1} E_\beta(\mathcal{R}_i^{\pi^k}(s_i^k))\delta_H^k.$$

**Step 3: Bound each term.** The first term can be bounded as

$$2(\exp(\beta(H+1)) - 1) \sum_{k=1}^{K} \sum_{h=1}^{H-1} c_h^k = 2(\exp(\beta(H+1)) - 1) \sum_{h=1}^{H-1} \sum_{k=1}^{K} \left( \sqrt{\frac{1}{2(N_h^k \vee 1)}\iota} + \sqrt{\frac{S}{(N_h^k \vee 1)}\iota} \right)$$

$$\leq 3(\exp(\beta(H+1)) - 1) \sum_{h=1}^{H-1} \sqrt{2S^2 AK\iota}$$

$$= 3(\exp(\beta(H+1)) - 1)\sqrt{2S^2 AK\iota}.$$

Observe that

$$\prod_{i=1}^{h} E_\beta(\mathcal{R}_i^{\pi^k}(s_i^k))\epsilon_h^k \in \exp(\beta h)D_h = \exp(\beta h)[1 - \exp(\beta(H+1-h)), \exp(\beta(H+1-h)) - 1]$$

$$\subseteq [1 - \exp(\beta(H+1)), \exp(\beta(H+1)) - 1],$$

thus we can bound the second term by Azuma-Hoeffding inequality: with probability at least $1 - \delta'$, the following holds

$$\sum_{k=1}^{K} \sum_{h=1}^{H-1} \prod_{i=1}^{h} E_\beta(\mathcal{R}_i^{\pi^k}(s_i^k))\epsilon_h^k \leq (\exp(\beta(H+1)) - 1)\sqrt{2KH\log(1/\delta')}.$$

727   We have

$$\sum_{k=1}^{K} \prod_{i=1}^{H-1} E_\beta(\mathcal{R}_i^{\pi^k}(s_i^k))\delta_H^k \le \sum_{k=1}^{K} \exp(\beta(H-1))L_1 c_H^k$$

$$\le \sum_{k=1}^{K} \exp(\beta(H-1))(\exp(\beta)-1)\left(\sqrt{\frac{1}{2(N_h^k \vee 1)}\iota} + \sqrt{\frac{S}{(N_h^k \vee 1)}\iota}\right)$$

$$\le 1.5(\exp(\beta H)-1)\sqrt{2S^2 AK\iota}.$$

728   Using a union bound and let $\delta = \delta' = \frac{\tilde\delta}{2}$, we have that with probability at least $1 - \delta$,

$$\text{Regret}(K) \le \frac{1}{\beta}\left(4.5(\exp(\beta(H+1))-1)\sqrt{2S^2 AK\iota} + (\exp(\beta(H+1))-1)\sqrt{2KH\iota}\right)$$

$$= \tilde{\mathcal{O}}\left(\frac{\exp(\beta H)-1}{\beta H}H\sqrt{HS^2 AT}\right),$$

729   where $\iota \triangleq \log(4SAT/\delta)$.

730   In contrast, if we use non-distributional analysis, we will arrive at

$$\text{Regret}(K) \le \tilde{\mathcal{O}}\left(\frac{\exp(\beta H)}{\beta}\sqrt{HS^2 AT}\right),$$

731   which blows up as $\beta \to 0$.

## D.3   Proof for regret lower bounds

733   **Notations.**   We define $\text{kl}(p,q) := p\log\frac{p}{q} + (1-p)\log\frac{1-p}{1-q}$ as the KL divergence between two
734   Bernoulli distributions with parameters $p$ and $q$.

### D.3.1   Correction of Lower Bound

736   [23] presents the following lower bound.

**Proposition 3** (Theorem 3,[23])**.** *For sufficiently large $K$ and $H$, the regret of any algorithm obeys*

$$\mathbb{E}[\text{Regret}(K)] \gtrsim \frac{e^{|\beta|H/2}-1}{|\beta|}\sqrt{T\log T}.$$

737   However, the lower bound itself and the proof are incorrect. The major mistake appears at the second
738   inequality of the following statements in their proof

$$\mathbb{E}[\text{Regret}(K)] \gtrsim \frac{\exp(\beta H/2)-1}{\beta}\sqrt{K\log(K)}$$

$$\gtrsim \frac{\exp(\beta H/2)-1}{\beta}\sqrt{KH\log(KH)}.$$

739   The authors establish the second inequality based on the following fact

740   **Fact 2** (Fact 5,[23] )**.** *For any $\alpha > 0$, the function $f_\alpha := \frac{e^{\alpha x}-1}{x}, x > 0$ is increasing and satisfies*
741   $\lim_{x\to 0} f_\alpha = \alpha$.

742   In fact, we can only use Fact 2 to derive $\frac{\exp(\beta H/2)-1}{\beta} \gtrsim H$, which combined with the first inequality
743   yields

$$\mathbb{E}[\text{Regret}(K)] \gtrsim H\sqrt{KH\log(KH)}.$$

744   It is a weaker lower bound and does not feature the dependence on $\beta$. The best result we can get
745   based on the original proof is that

**Proposition 4** (Correction of Theorem 3,[23] )**.** *For sufficiently large $K$ and $H$, the regret of any
algorithm obeys*

$$\mathbb{E}[\text{Regret}(K)] \gtrsim \frac{e^{|\beta|H/2}-1}{|\beta|}\sqrt{K\log K}.$$

### D.3.2 Proof of Theorem 3

We introduce some notations here. We define the probability measure induced by an algorithm $\mathscr{A}$ and an MDP instance $\mathcal{M}$ as

$$\mathbb{P}_{\mathscr{A}\mathcal{M}}(\mathcal{F}^{K+1}) := \prod_{k=1}^{K} \mathbb{P}_{\mathscr{A}_k(\mathcal{F}^k)\mathcal{M}}(\mathcal{I}_H^k | s_1^k),$$

where $\mathbb{P}_{\pi\mathcal{M}}$ is the probability measure induced by a policy $\pi$ and $\mathcal{M}$, which is defined as

$$\mathbb{P}_{\pi\mathcal{M}}(\mathcal{I}_H | s_1) := \prod_{h=1}^{H} \pi_h(a_h | s_h) P_h^{\mathcal{M}}(s_{h+1} | s_h, a_h).$$

Note that the probability measure for the truncated history $\mathcal{H}_h^k$ can be obtained by marginalization

$$\mathbb{P}_{\mathscr{A}\mathcal{M}}(\mathcal{H}_h^k) = \mathbb{P}_{\mathscr{A}\mathcal{M}}(\mathcal{F}^k) \mathbb{P}_{\mathscr{A}_k(\mathcal{F}^k)\mathcal{M}}(\mathcal{I}_h^k).$$

We denote by $\mathbb{P}_{\mathscr{A}\mathcal{M}}$ and $\mathbb{E}_{\mathscr{A}\mathcal{M}}$ the probability measure and expectation induced by $\mathscr{A}$ and $\mathcal{M}$. For the sake of simplicity, the dependency on $\mathscr{A}$ and $\mathcal{M}$ may be dropped if it is clear in the context.

**Fact 3** (Lemma 1, [26]). *Consider a measurable space $(\Omega, \mathcal{F})$ equipped with two distributions $\mathbb{P}_1$ and $\mathbb{P}_2$. For any $\mathcal{F}$-measurable function $Z : \Omega \to [0, 1]$, we have*

$$\mathrm{KL}\left(\mathbb{P}_1, \mathbb{P}_2\right) \geq \mathrm{kl}\left(\mathbb{E}_1[Z], \mathbb{E}_2[Z]\right),$$

*where $\mathbb{E}_1$ and $\mathbb{E}_2$ are the expectations under $\mathbb{P}_1$ and $\mathbb{P}_2$ respectively.*

**Fact 4** (Lemma 5, [20]). *Let $\mathcal{M}$ and $\mathcal{M}'$ be two MDPs that are identical except for their transition probabilities, denoted by $P_h$ and $P_h'$, respectively. Assume that we have $\forall (s, a)$, $P_h(\cdot \mid s, a) \ll P_h'(\cdot \mid s, a)$. Then, for any stopping time $\tau$ with respect to $(I_k)_{k \geq 1}$ that satisfies $\mathbb{P}_{\mathcal{M}}[\tau < \infty] = 1$*

$$\mathrm{KL}\left(\mathbb{P}_{\mathcal{M}}, \mathbb{P}_{\mathcal{M}'}\right) = \sum_{(s,a,h) \in \mathcal{S} \times \mathcal{A} \times [H-1]} \mathbb{E}_{\mathcal{M}}\left[N_h^\tau(s, a)\right] \mathrm{KL}\left(P_h(\cdot \mid s, a), P_h'(\cdot \mid s, a)\right).$$

**Lemma 14.** *If $\epsilon \geq 0$, $p \geq 0$ and $p + \epsilon \in [0, \frac{1}{2}]$, then $\mathrm{kl}(p, p + \epsilon) \leq \frac{\epsilon^2}{2p(1-p)} \leq \frac{\epsilon^2}{p}$.*

*Proof.* Fix $q \in [0, 1]$, let $h(p) := \mathrm{kl}(p, q)$. It is immediate that

$$h'(p) = \log \frac{p}{q} - \log \frac{1-p}{1-q},$$

$$h''(p) = \frac{1}{p(1-p)} > 0.$$

Therefore $h(p)$ is strictly convex, increasing in $(q, 1)$ and decreasing in $(0, q)$. By Taylor's expansion, we have that

$$h(p) = h(q) + h'(q)(p - q) + \frac{1}{2} h''(r)(p - q)^2 = \frac{(p - q)^2}{2r(1 - r)}$$

for some $r \in [p, q]$ $(p < q)$ or $r \in [q, p]$ $(p > q)$. In particular, for any $\epsilon \geq 0$ such that $q = p + \epsilon \leq \frac{1}{2}$ it follows that

$$\mathrm{kl}(p, p + \epsilon) = \frac{(p - q)^2}{2r(1 - r)}\Big|_{q = p + \epsilon} = \frac{\epsilon^2}{2r(1 - r)} \leq \frac{\epsilon^2}{2p(1 - p)} \leq \frac{\epsilon^2}{p},$$

where the first inequality follows from the fact that $r \mapsto r(1 - r)$ is increasing in $[p, p + \epsilon] \subset [0, \frac{1}{2}]$ and the second inequality is due to that $1 - p \geq \frac{1}{2}$. $\qquad\square$

The proof of Theorem 3 adopts the same construction of hard MDP class $\mathcal{C}$ as [20].

*Proof.* We consider the case that $\beta > 0$. Fix an arbitrary algorithm $\mathscr{A}$. We introduce three types of special states for the hard MDP class: a waiting state $s_w$ where the agent starts and may stay until stage $\bar{H}$, after that it has to leave; a good state $s_g$ which is absorbing and is the only rewarding state; a bad state $s_b$ that is absorbing and provides no reward. The rest $S - 3$ states are part of a $A$-ary tree

of depth $d-1$. The agent can only arrive $s_w$ from the root node $s_{root}$ and can only reach $s_g$ and $s_b$ from the leaves of the tree.

Let $\bar{H} \in [H-d]$ be the first parameter of the MDP class. We define $\tilde{H} := \bar{H} + d + 1$ and $H' := H + 1 - \tilde{H}$. We denote by $\mathcal{L} := \{s_1, s_2, ..., s_{\bar{L}}\}$ the set of $\bar{L}$ leaves of the tree. For each $u^* := (h^*, \ell^*, a^*) \in [d+1 : \bar{H}+d] \times \mathcal{L} \times \mathcal{A}$, we define an MDP $\mathcal{M}_{u^*}$ as follows. The transitions in the tree are deterministic, hence taking action $a$ in state $s$ results in the $a$-th child of node $s$. The transitions from $s_w$ are defined as

$$P_h(s_w \mid s_w, a) := \mathbb{I}\{a = a_w, h \le \bar{H}\} \quad \text{and} \quad P_h(s_{root} \mid s_w, a) := 1 - P_h(s_w \mid s_w, a).$$

The transitions from any leaf $s_i \in \mathcal{L}$ are specified as

$$P_h(s_g \mid s_i, a) := p + \Delta_{u^*}(h, s_i, a) \quad \text{and} \quad P_h(s_b \mid s_i, a) := p - \Delta_{u^*}(h, s_i, a),$$

where $\Delta_{u^*}(h, s_i, a) := \epsilon \mathbb{I}\{(h, s_i, a) = (h^*, s_{\ell^*}, a^*)\}$ for some constants $p \in [0,1]$ and $\epsilon \in [0, \min(1-p, p)]$ to be determined later. $p$ and $\epsilon$ are the second and third parameters of the MDP class. Observe that $s_g$ and $s_b$ are absorbing, therefore we have $\forall a, P_h(s_g \mid s_g, a) := P_h(s_b \mid s_b, a) := 1$. The reward is a deterministic function of the state

$$r_h(s, a) := \mathbb{I}\{s = s_g, h \ge \tilde{H}\}.$$

Finally we define a reference MDP $\mathcal{M}_0$ which differs from the previous MDP instances only in that $\Delta_0(h, s_i, a) := 0$ for all $(h, s_i, a)$. For each $\epsilon, p$ and $\bar{H}$, we define the MDP class

$$\mathcal{C}_{\bar{H}, p, \epsilon} := \mathcal{M}_0 \cup \{\mathcal{M}_{u^*}\}_{u^* \in [d+1:\bar{H}+d] \times \mathcal{L} \times \mathcal{A}}.$$

The total expected ERM value of $\mathscr{A}$ is given by

$$\mathbb{E}_{\mathscr{A}, \mathcal{M}_{u^*}} \left[ \sum_{k=1}^{K} U_\beta \left( \sum_{h=1}^{H} r_h(s_h^k, a_h^k) | \pi^k \right) \right]$$

$$= \mathbb{E}_{\mathscr{A}, \mathcal{M}_{u^*}} \left[ \sum_{k=1}^{K} \frac{1}{\beta} \log \mathbb{E}_{\mathscr{A}, \mathcal{M}_{u^*}} \left[ \exp \left( \beta \sum_{h=1}^{H} r_h(s_h^k, a_h^k) \right) \right] \right]$$

$$= \mathbb{E}_{\mathscr{A}, \mathcal{M}_{u^*}} \left[ \sum_{k=1}^{K} \frac{1}{\beta} \log \mathbb{E}_{\pi^k, \mathcal{M}_{u^*}} \left[ \exp \left( \beta \sum_{h=\tilde{H}}^{H} \mathbb{I}\{s_h^k = s_g\} \right) \right] \right]$$

$$= \mathbb{E}_{\mathscr{A}, \mathcal{M}_{u^*}} \left[ \sum_{k=1}^{K} \frac{1}{\beta} \log \mathbb{E}_{\pi^k, \mathcal{M}_{u^*}} \left[ \exp(\beta H' \mathbb{I}\{s_{\tilde{H}}^k = s_g\}) \right] \right]$$

$$= \mathbb{E}_{\mathscr{A}, \mathcal{M}_{u^*}} \left[ \sum_{k=1}^{K} \frac{1}{\beta} \log(\exp(\beta H') \mathbb{P}_{\pi^k, \mathcal{M}_{u^*}}(s_{\tilde{H}}^k = s_g) + \mathbb{P}_{\pi^k, \mathcal{M}_{u^*}}(s_{\tilde{H}}^k = s_b)) \right],$$

where the second equality follows from the fact that the reward is non-zero only after step $\tilde{H}$, the third equality is due to that the agent gets into absorbing state when $h \ge \tilde{H}$. Define $x_h^k := (s_h^k, a_h^k)$ for each $(k, h)$ and $x^* := (s_{\ell^*}, a^*)$, then it is not hard to obtain that

$$\mathbb{P}_{\pi^k, u^*}\left[ s_{\tilde{H}}^k = s_g \right] = \sum_{h=1+d}^{\bar{H}+d} p \mathbb{P}_{\pi^k, u^*}\left( s_h^k \in \mathcal{L} \right) + \mathbb{I}\{h = h^*\} \mathbb{P}_{\pi^k, u^*}(x_h^k = x^*)\varepsilon$$

$$= p + \epsilon \mathbb{P}_{\pi^k, u^*}(x_{h^*}^k = x^*).$$

For an MDP $\mathcal{M}_{u^*}$, the optimal policy $\pi^{*, \mathcal{M}_{u^*}}$ starts to traverse the tree at step $h^* - d$ then chooses to reach the leaf $s_{l^*}$ and performs action $a^*$. The corresponding optimal value in any of the MDPs is $V^{*, \mathcal{M}_{u^*}} = \frac{1}{\beta} \log(\exp(\beta H')(p + \epsilon) + 1 - p - \epsilon)$. Define $p_{u^*}^k := \mathbb{P}_{\pi^k, u^*}(x_{h^*}^k = x^*)$, then the

expected regret of $\mathscr{A}$ in $\mathcal{M}_{u^*}$ can be bounded below as

$$\mathbb{E}_{\mathscr{A},\mathcal{M}_{u^*}}\left[\mathrm{Regret}(\mathscr{A},\mathcal{M}_{u^*},K)\right]$$

$$= \mathbb{E}_{\mathscr{A},\mathcal{M}_{u^*}}\left[\sum_{k=1}^K V^{*,\mathcal{M}_{u^*}} - U_\beta\left(\sum_{h=1}^H r_h(x_h^k)|\pi^k\right)\right]$$

$$= \mathbb{E}_{\mathscr{A},\mathcal{M}_{u^*}}\left[\sum_{k=1}^K \frac{1}{\beta}\log\frac{\exp(\beta H')(p+\epsilon)+1-p-\epsilon}{\exp(\beta H')(p+\epsilon p_{u^*}^k)+1-p-\epsilon p_{u^*}^k}\right]$$

$$= \mathbb{E}_{\mathscr{A},\mathcal{M}_{u^*}}\left[\sum_{k=1}^K \frac{1}{\beta}\log\left(1+\frac{\epsilon(1-p_{u^*}^k)(\exp(\beta H')-1)}{\exp(\beta H')(p+\epsilon p_{u^*}^k)+1-p-\epsilon p_{u^*}^k}\right)\right]$$

$$\geq \mathbb{E}_{\mathscr{A},\mathcal{M}_{u^*}}\left[\sum_{k=1}^K \frac{1}{\beta}\log\left(1+\frac{\epsilon(1-p_{u^*}^k)(\exp(\beta H')-1)}{1+1}\right)\right]$$

$$\geq \mathbb{E}_{\mathscr{A},\mathcal{M}_{u^*}}\left[\frac{\exp(\beta H')-1}{4\beta}\epsilon\sum_{k=1}^K(1-p_{u^*}^k)\right]$$

$$= \frac{\exp(\beta H')-1}{4\beta}\epsilon\sum_{k=1}^K(1-\mathbb{E}_{\mathscr{A},\mathcal{M}_{u^*}}[p_{u^*}^k])$$

$$= \frac{\exp(\beta H')-1}{4\beta}K\epsilon\left(1-\frac{1}{K}\mathbb{E}_{\mathscr{A},\mathcal{M}_{u^*}}[N_K(u^*)]\right).$$

The first inequality holds by setting $p+\epsilon \leq \exp(-\beta H')$. The second inequality holds by letting $\epsilon \leq 2\exp(-\beta H')$ since $\log(1+x) \geq \frac{x}{2}$ for $x \in [0,1]$. The last equality follows from the fact that

$$\mathbb{E}_{\mathscr{A},\mathcal{M}_{u^*}}[p_{u^*}^k] = \mathbb{E}_{\mathscr{A},\mathcal{M}_{u^*}}[\mathbb{P}_{\pi^k,u^*}(x_{h^*}^k=x^*)] = \mathbb{P}_{\mathscr{A},u^*}(x_{h^*}^k=x^*) = \mathbb{E}_{\mathscr{A},u^*}[\mathbb{I}\{(x_{h^*}^k=x^*)\}]$$

and the definition of $N_K(u^*) := \sum_{k=1}^K \mathbb{I}\{x_{h^*}^k=x^*\}$.

The maximum of the regret can be bounded below by the mean over all instances as

$$\max_{u^*\in[d+1:\bar{H}+d]\times\mathcal{L}\times\mathcal{A}}\mathrm{Regret}(\mathscr{A},\mathcal{M}_{u^*},K) \geq \frac{1}{\bar{H}\bar{L}A}\sum_{u^*\in[d+1:\bar{H}+d]\times\mathcal{L}\times\mathcal{A}}\mathrm{Regret}(\mathscr{A},\mathcal{M}_{u^*},K)$$

$$\geq \frac{\exp(\beta H')-1}{4\beta}K\epsilon\left(1-\frac{1}{\bar{L}AK\bar{H}}\sum_{u^*\in[d+1:\bar{H}+d]\times\mathcal{L}\times\mathcal{A}}\mathbb{E}_{u^*}[N_K(u^*)]\right).$$

Observe that it can be further bounded if we can obtain an upper bound on $\sum_{u^*\in[d+1:\bar{H}+d]\times\mathcal{L}\times\mathcal{A}}\mathbb{E}_{u^*}[N_K(u^*)]$, which can be done by relating each expectation to the expectation under the reference MDP $\mathcal{M}_0$.

By applying Fact 3 with $Z = \frac{N_K(u^*)}{K} \in [0,1]$, we have

$$\mathrm{kl}\left(\frac{1}{K}\mathbb{E}_0[N_K(u^*)],\frac{1}{K}\mathbb{E}_{u^*}[N_K(u^*)]\right) \leq \mathrm{KL}(\mathbb{P}_0,\mathbb{P}_{u^*}).$$

By Pinsker's inequality, it implies that

$$\frac{1}{K}\mathbb{E}_{u^*}[N_K(u^*)] \leq \frac{1}{K}\mathbb{E}_0[N_K(u^*)] + \sqrt{\frac{1}{2}\mathrm{KL}(\mathbb{P}_0,\mathbb{P}_{u^*})}.$$

Since $\mathcal{M}_0$ and $\mathcal{M}_{u^*}$ only differs at stage $h^*$ when $(s,a)=x^*$, it follows from Fact 4 that

$$\mathrm{KL}(\mathbb{P}_0,\mathbb{P}_{u^*}) = \mathbb{E}_0[N_K(u^*)]\mathrm{kl}(p,p+\varepsilon).$$

By Lemma 14, we have $\mathrm{kl}(p, p+\epsilon) \leq \frac{\epsilon^2}{p}$ for $\epsilon \geq 0$ and $p + \epsilon \in [0, \frac{1}{2}]$. Consequently,

$$\frac{1}{K} \sum_{u^* \in [d+1:\bar{H}+d] \times \mathcal{L} \times \mathcal{A}} \mathbb{E}_{u^*}[N_K(u^*)]$$

$$\leq \frac{1}{K} \mathbb{E}_0 \left[ \sum_{u^* \in [d+1:\bar{H}+d] \times \mathcal{L} \times \mathcal{A}} N_K(u^*) \right] + \frac{\epsilon}{\sqrt{2p}} \sum_{u^* \in [d+1:\bar{H}+d] \times \mathcal{L} \times \mathcal{A}} \sqrt{\mathbb{E}_0[N_K(u^*)]}$$

$$\leq 1 + \frac{\epsilon}{\sqrt{2p}} \sqrt{\bar{L}AK\bar{H}},$$

where the second inequality is due to the Cauchy-Schwartz inequality and that $\sum_{u^* \in [d+1:\bar{H}+d] \times \mathcal{L} \times \mathcal{A}} N_K(u^*) = K$.
It follows that

$$\max_{u^* \in [d+1:\bar{H}+d] \times \mathcal{L} \times \mathcal{A}} \mathrm{Regret}(\mathscr{A}, \mathcal{M}_{u^*}, K) \geq \frac{\exp(\beta H') - 1}{4\beta} K\epsilon \left( 1 - \frac{1}{\bar{L}A\bar{H}} - \frac{\frac{\epsilon}{\sqrt{2p}}\sqrt{\bar{L}AK\bar{H}}}{\bar{L}A\bar{H}} \right).$$

Choosing $\epsilon = \sqrt{\frac{p}{2}}(1 - \frac{1}{\bar{L}A\bar{H}})\sqrt{\frac{\bar{L}A\bar{H}}{K}}$ maximizes the lower bound

$$\max_{u^* \in [d+1:\bar{H}+d] \times \mathcal{L} \times \mathcal{A}} \mathrm{Regret}(\mathscr{A}, \mathcal{M}_{u^*}, K) \geq \frac{\sqrt{p}}{8\sqrt{2}} \frac{\exp(\beta H') - 1}{\beta} \left( 1 - \frac{1}{\bar{L}A\bar{H}} \right)^2 \sqrt{\bar{L}AK\bar{H}}.$$

Since $S \geq 6$ and $A \geq 2$, we have $\bar{L} = (1 - \frac{1}{A})(S-3) + \frac{1}{A} \geq \frac{S}{4}$ and $1 - \frac{1}{\bar{L}A\bar{H}} \geq 1 - \frac{1}{\frac{6}{4} \cdot 2} = \frac{2}{3}$. Choose $\bar{H} = \frac{H}{3}$ and use the assumption that $d \leq \frac{H}{3}$ to obtain that $H' = H - d - \bar{H} \geq \frac{H}{3}$. Now we choose $p = \frac{1}{4}\exp(-\beta H')$ and $\epsilon = \sqrt{\frac{p}{2}}(1 - \frac{1}{\bar{L}A\bar{H}})\sqrt{\frac{\bar{L}A\bar{H}}{K}} \leq \frac{1}{2\sqrt{2}}\exp(-\beta H'/2)\sqrt{\frac{\bar{L}A\bar{H}}{K}} \leq \frac{1}{4}\exp(-\beta H')$ if $K \geq 2\exp(\beta H')\bar{L}A\bar{H}$. Such choice of $p$ and $\epsilon$ guarantees the assumption of Lemma 14 and that $p + \epsilon \leq \exp(-\beta H')$, $\epsilon \leq 2\exp(-\beta H')$. Finally we use the fact that $\sqrt{\bar{L}AK\bar{H}} \geq \frac{1}{2\sqrt{3}}\sqrt{SAKH}$ to obtain

$$\max_{u^* \in [d+1:\bar{H}+d] \times \mathcal{L} \times \mathcal{A}} \mathrm{Regret}(\mathscr{A}, \mathcal{M}_{u^*}, K) \geq \frac{1}{72\sqrt{6}} \frac{\exp(\beta H/6) - 1}{\beta} \sqrt{SAKH}.$$

$\square$

# E  Proof for Propositions

For notational simplicity, we write $\pi_{h_1:h_2} = \{\pi_{h_1}, \pi_{h_1+1}, \cdots, \pi_{h_2}\}$ for two positive integers $h_1 < h_2 \leq H$.

*Proof of Proposition 1.* Notice that there exists some optimal policy for sub-problems at each step, which will be shown in Proposition 2. Suppose that the truncated policy $\pi^*_{h:H}$ is not optimal for this subproblem, then there exists an optimal policy $\tilde{\pi}_{h:H}$ such that

$$\exists \tilde{s}_h \quad \text{occurring with positive probability,} \quad V_h^{\tilde{\pi}_{h:H}}(\tilde{s}_h) > V_h^{\pi^*_{h:H}}(\tilde{s}_h).$$

There exists a state $\tilde{s}_{h-1}$ which occurs with positive probability and $P_{h-1}(\tilde{s}_h | \tilde{s}_{h-1}, \pi^*_{h-1}(\tilde{s}_{h-1})) > 0$ such that

$$U_\beta(\nu_{h-1}^{\pi^*_{h-1}, \tilde{\pi}_{h:H}}(\tilde{s}_{h-1})) = U_\beta(\mathcal{R}_{h-1}(\tilde{s}_{h-1}, \pi^*_{h-1}(\tilde{s}_{h-1}))) + U_\beta\left( \left[ P_{h-1}\nu_h^{\tilde{\pi}_{h:H}} \right] (\tilde{s}_{h-1}, \pi^*_{h-1}(\tilde{s}_{h-1})) \right)$$

$$> U_\beta(\mathcal{R}_{h-1}(\tilde{s}_{h-1}, \pi^*_{h-1}(\tilde{s}_{h-1}))) + U_\beta\left( \left[ P_{h-1}\nu_h^{\pi^*_{h:H}} \right] (\tilde{s}_{h-1}, \pi^*_{h-1}(\tilde{s}_{h-1})) \right)$$

$$= U_\beta\left( \nu_{h-1}^{\pi^*_{h-1:H}}(\tilde{s}_{h-1}) \right),$$

where the inequality is due to the strict monotonicity preserving property of $U_\beta$. It follows that $\{\pi^*_{h-1}, \tilde{\pi}_h, , ..., \tilde{\pi}_H\}$ is a strictly better policy than $\{\pi^*_{h-1}, \pi^*_h, , ..., \pi^*_H\}$ for the subproblem from

818   $h - 1$ to $H$. Suppose for $h' + 1 \in [2, h-1]$, $\{\pi^*_{h'+1}, ..., \tilde{\pi}_h, , ..., \tilde{\pi}_H\}$ is a strictly better policy
819   than $\{\pi^*_{h'+1}, ..., \pi^*_h, , ..., \pi^*_H\}$ for the sub-problem from $h' + 1$ to $H$. Similarly we can obtain
820   that $\{\pi^*_{h'}, ..., \tilde{\pi}_h, , ..., \tilde{\pi}_H\}$ is also a strictly better policy than $\{\pi^*_{h'}, ..., \pi^*_h, , ..., \pi^*_H\}$. Repeating the
821   above arguments finally yields that $\{\pi^*_1, \pi^*_2, ..., \tilde{\pi}_h, , ..., \tilde{\pi}_H\}$ is a strictly better policy than $\pi^* =$
822   $\{\pi^*_1, \pi^*_2, ..., \pi^*_H\}$. This is contradicted to the assumption that $\pi^* = \{\pi^*_1, \pi^*_2, ..., \pi^*_H\}$ is an optimal
823   policy. $\qquad\square$

*Proof of Proposition 2.* Throughout the proof we drop the dependence on $*$ for the ease of notation.
The proof follows from induction. Notice that by distributional Bellman equation, $\eta_h(s_h)$ and $V_h(s_h)$
are the return distribution at state $s_h$ at step $h$ following policy $\pi_{h:H}$ and value function respectively.
At step $H$, it is obvious that $\pi_H$ is the optimal policy that maximizes the ERM value at the final
step for each state $s_H \in \mathcal{S}$. Now fix $h \in [H-1]$, assume that $\pi_{h+1:H}$ is the optimal policy for the
subproblem

$$V_{h+1}^{\pi_{h+1:H}}(s_{h+1}) = \max_{\pi'_{h+1:H}} V_{h+1}^{\pi'_{h+1:H}}(s_{h+1}), \forall s_{h+1}.$$

824   In other words,

$$U_\beta(\nu_{h+1}(s_{h+1})) = U_\beta(\nu_{h+1}^{\pi_{h+1:H}}(s_{h+1})) = \max_{\pi'_{h+1:H}} U_\beta(\nu_{h+1}^{\pi'_{h+1:H}}(s_{h+1}))$$

$$\geq U_\beta(\nu_{h+1}^{\pi'_{h+1:H}}(s_{h+1})), \forall \pi'_{h+1:H}, \forall s_{h+1}.$$

825   It follows that $\forall s_h$,

$$V_h(s_h) = Q_h(s_h, \pi_h(s_h)) = U_\beta(\nu_h^{\pi_{h:H}}(s_h)) = \max_{a_h} U_\beta(\eta_h(s_h, a_h))$$

$$= \max_{a_h} \{U_\beta(\mathcal{R}_h(s_h, a_h)) + U_\beta([P_h \nu_{h+1}](s_h, a_h))\}$$

$$\geq \max_{a_h} \left\{ U_\beta(\mathcal{R}_h(s_h, a_h)) + \max_{\pi'_{h+1:H}} U_\beta\left(\left[P_h \nu_{h+1}^{\pi'_{h+1:H}}\right](s_h, a_h)\right) \right\}$$

$$= \max_{\pi'_h} \left\{ U_\beta(\mathcal{R}_h(s_h, \pi'_h(s_h))) + \max_{\pi'_{h+1:H}} U_\beta\left(\left[P_h \nu_{h+1}^{\pi'_{h+1:H}}\right](s_h, a_h)\right) \right\}$$

$$= \max_{\pi'_{h:H}} \left\{ U_\beta(\mathcal{R}_h(s_h, \pi'_h(s_h))) + U_\beta\left(\left[P_h \nu_{h+1}^{\pi'_{h+1:H}}\right](s_h, \pi'_h(s_h))\right) \right\}$$

$$= \max_{\pi'_{h:H}} U_\beta\left(\nu_h^{\pi'_{h+1:H}}(s_h)\right).$$

826   Hence $V_h$ is the optimal value function at step $h$ and $\pi_{h:H}$ is the optimal policy for the sub-problem
827   from $h$ to $H$. The induction is completed. $\qquad\square$

828   **Definition 1.** *For two algorithms $\mathscr{A}$ and $\tilde{\mathscr{A}}$, we say that $\mathscr{A}$ is equivalent to $\tilde{\mathscr{A}}$ (vice versa) if for any*
829   *$k \in [K]$, any $\mathcal{F}_k$ it holds that $\mathscr{A}(\mathcal{F}_k) = \tilde{\mathscr{A}}(\mathcal{F}_k)$.*

830   It follows from the induction that the whole history/trajectory $\mathcal{F}_{K+1}$ generated by the interaction
831   between each of two equivalent algorithms and any MDP instance follows the same distribution.
832   Moreover, the two algorithms possess equal regret.

833   **Proposition 5** (Equivalence between `ROVI` and `RODI-MB`)**.** *Algorithm 5 (Algorithm 2) is equivalent*
834   *to Algorithm 6 (Algorithm 3).*

835   *Proof.* We only prove the case that $\beta > 0$. The case that $\beta < 0$ follows analogously. Fix an
836   arbitrary $k \in [K]$ and $\mathcal{F}_k = \{s_1^1, a_1^1, R_1^1, \cdots, s_H^{k-1}, a_H^{k-1}, R_H^{k-1}\}$. Denote by $\mathscr{A}$ ($\tilde{\mathscr{A}}$) and $\{\pi_h^k\}$
837   ($\{\tilde{\pi}_h^k\}$) Algorithm 6 (Algorithm 5) and the associated policy sequence. It suffices to prove that $\pi^k$
838   coincides with $\tilde{\pi}^k$ for the same history $\mathcal{F}_k$. By the definition of the two algorithms, we have

$$\tilde{\pi}_h^k(s) = \arg\max_a Q_h^k(s, a) = U_\beta(\eta_h^k(s, a)), \ \pi_h^k(s) = \arg\max_a J_h^k(s, a).$$

839   If $J_h^k(s, a) = E_\beta(\eta_h^k(s, a)) = \exp(\beta Q_h^k(s, a))$ for any $(s, a)$, then $\pi_h^k = \tilde{\pi}_h^k$ due to the monotonicity
840   of the exponential function. We will prove that $J_h^k(s, a) = E_\beta(\eta_h^k(s, a))$ by the induction. Notice

841 that $J_H^k(s, a) = E_\beta(\eta_H^k(s, a))$. Assume that $J_h^k(s, a) = E_\beta(\eta_h^k(s, a))$ for some $h \in [H]$. It follows
842 that $\pi_h^k = \tilde{\pi}_h^k$ and

$$W_h^k(s) = \max_a J_h^k(s, a) = J_h^k(s, \pi_h^k(s)) = E_\beta(\eta_h^k(s, \pi_h^k(s))) = E_\beta(\eta_h^k(s, \tilde{\pi}_h^k(s)))$$
$$= E_\beta(\nu_h^k(s)).$$

843 Given the same history $\mathcal{F}_k$, the two algorithms share the empirical transition model $\hat{P}_{h-1}^k$, the
844 empirical reward distribution $\hat{\mathcal{R}}_{h-1}^k$, the count $N_{h-1}^k$, and the optimism constants $c_{h-1,1}^k$, $c_{h-1,2}^k$.
845 Therefore they also share the optimistic transition model $\tilde{P}_{h-1}^k$ as well as the optimistic reward
846 distribution $\tilde{\mathcal{R}}_{h-1}^k$. According to the update formula of Algorithm 6, we have that for any $(s, a)$

$$J_{h-1}^k(s, a) = E_\beta\left(\tilde{\mathcal{R}}_{h-1}^k(s, a)\right)\left[\tilde{P}_{h-1}^k W_h^k\right](s, a) = E_\beta\left(\tilde{\mathcal{R}}_{h-1}^k(s, a)\right) E_\beta\left(\left[\tilde{P}_{h-1}^k \nu_h^k\right](s, a)\right)$$
$$= E_\beta\left(\left[\mathcal{B}(\tilde{P}_{h-1}^k, \tilde{\mathcal{R}}_{h-1}^k)\nu_h^k\right](s, a)\right)$$
$$= E_\beta\left(\eta_{h-1}^k(s, a)\right).$$

847 Thus the proof for the case of random reward is completed. The proof for the case of deterministic
848 reward follows analogously.

849 $\qquad\qquad\qquad\qquad\qquad\qquad\qquad\qquad\qquad\qquad\qquad\qquad\qquad\qquad\qquad\qquad\qquad\qquad\qquad\quad$ $\square$