# OpenReview forum: "Near-optimal Distributional Reinforcement Learning towards Risk-sensitive Control"
_NeurIPS.cc/2022/Conference — NeurIPS 2022 Submitted_

### Official Review · Reviewer_SdUB · 2022-07-11

**Rating:** 5
**Confidence:** 3
**Soundness:** 3 good
**Presentation:** 3 good
**Contribution:** 3 good

**Summary:**

This work proposes to build a distributional dynamic programming framework to optimize the entropic risk measure of the return. Under the new framework, both model-based and model-free algorithms are proposed, which achieve regret bound that matches one of the existing works in risk sensitive control. The lower bound of rish sensitive control problem is also improved using the new framework.

**Questions:**

1. In Appendix A of [22], they also provide connections between risk-sensitive RL and distributional RL through the exponential Bellman equation. How is the proposed framework related to theirs? It would be helpful if there's a section dedicated to explain the relation and potential difference w.r.t. their analysis.

2. If Alg. 2 is actually equivalent to Alg. 3, what is the benefit of distributional framework over existing risk sensitive control approaches? E.g., what are the cases when the proposed framework is more favourable, either theoretically or empirically?

3. Does the improvement in the lower bound in Section 5.3 come from the newly proposed framework, or just different proof techniques?

**Limitations:**

The limitation of the proposed method are discussed, but not adequately.

**Strengths And Weaknesses:**

Strengths:

1. The proposed distributional dynamics programming framework is novel to my best knowledge. Differences in the analysis between distributional RL and previous work [22] are discussed, which shows that the distributional analysis can avoid the explode factor and shows the significance of the proposed framework in theory.

2. The presentation of the paper is clear. Key steps in the analysis are highlighted, with detailed discussion on how they would affect the final regret bound. Algorithm designs are also explained in detail.

Weaknesses:

1. Since this work is highly relevant to [22], the significance of the work can benefit from a more comprehensive comparison  w.r.t. to previous work [22]; See Questions 1 and 2. Also, further discussion on another relevant work [1] can also improve the significance.

2. The novelty of the proposed algorithms can benefit from further discussion. The equivalence of the proposed model-based algorithm to ROVI is discussed, which raises a natural question: if distributional RL is somewhat equivalent to the existing algorithms, existing analysis could also apply to DRL, then is the claim in line 10 "this is the first regret analysis of DRL" still valid?

[22] Yingjie Fei, Zhuoran Yang, Yudong Chen, and Zhaoran Wang. Exponential bellman equation and improved regret bounds for risk-sensitive reinforcement learning. Advances in Neural Information Processing Systems, 34, 2021.

---

> ### Author Response · Authors · 2022-08-02
> **Response to Reviewer SdUB**
>
> Thank you for the constructive feedback. We would like to address your concerns as follows.
> ***
> **W1** ''comparison with [22] and [1]''
> **A**:  Thank you for your suggestions. We will add the comparisons (see below) in the revised version. The comparison with [22] is closely related to **Q1**, so we leave it to the response to **Q1**. Although we share the same term ``risk-sensitive DDP framework'' with [1], our framework differs from theirs in several aspects. We summarize the differences between our work and [1] as follows.
> 1. Setting. [1] considers the discounted MDP with infinite horizon, but we consider the episodic MDP setting.   Moreover, [1] assumes that the model is known, while we propose DRL algorithms when the model is unknown (i.e., the learning). Neither RL algorithms suitable for unknown model nor sample complexity guarantee is provided in their work.
> 2. Risk measure. [1] establish the risk-sensitive DDP framework using the risk measure Conditional Value at Risk, while our work considers the entropic risk measure.
> ***
> **W2/Q2** `If Alg.2 is actually equivalent to Alg.3, what is the benefit of the distributional framework over existing risk-sensitive control approaches''
> **A**: We would like to clarify a misunderstanding of the concept ''classical algorithm'' in line 186, where we write ''can be reduced to a *classical/non-distributional* reinforcement learning algorithm''. Here the ''classical reinforcement learning algorithm'' does not mean the traditional RL algorithm that is well known or already exists in the literature. Alternatively, we refer to it as the non-distributional reinforcement learning algorithm, which deals with the one-dimensional values instead of the whole distributions.
> We realize the word ''classical'' is misleading, so we decide not to use it and only use the term ''non-distributional''. Meanwhile, the non-distributional algorithm ROVI is also a novel model-based algorithm that does not exists in the literature. It is tempting to use the standard analysis that does not involve distributions to derive the regret upper bound of Alg.3. However, we have mentioned in Section 5.1 (line 257-line 262) that the non-distributional analysis induces a factor of $\frac{1}{|\beta|}\exp(|\beta| H)$, which explodes as $|\beta|\rightarrow0$. We fix it by invoking a distributional analysis to the DRL algorithm (Alg.3). Hence the distributional framework is indispensable for getting meaningful regret upper bound. Note that the distributional analysis for Alg.3 is novel compared with the standard analysis. Thus, the claim in line 10 ''this is the first regret analysis of DRL" still holds.
> ***
> **Q1** ''How is the proposed framework related to  Appendix A of [22]''
> **A**: In Appendix A of [22], the authors show that one can estimate the moment generating function (MGF) based on the exponential Bellman equation by varying the value of $\beta$
> $$ \mathbb{E}[\exp (\beta \cdot Z_{h}^{\pi}(s, a)] = e^{\beta \cdot Q_{h}^{\pi}(s, a)}=\mathbb{E}_{X'}[\text{exp}( \beta ( r_h(s,a) + V _{h+1}^{\pi} (X') ) ) ]. $$
> Notice that here $\beta$ is not considered a fixed number but a variable of the MGF of return $Z^{\pi}_h(s, a)$. Besides, the equation holds for a fixed policy $\pi$. One can get the MGF of $Z^{\pi}_h(s,a)$ evaluated at $\beta$ by solving the exponential Bellman equation for a particular value $\beta$. By varying all possible values of $\beta$, one can recover the MGF of $Z^{\pi}_h(s,a)$. Since an MGF uniquely identifies a distribution, one can also recover the distribution of $Z^{\pi}_h(s, a)$ in principle. Below are the differences with our risk-sensitive distributional dynamic programming framework.
> 1. Direction of connection. The connection established in [22] is ''RSRL $\rightarrow$ DRL'', while our framework establish the connection ''DRL$\rightarrow$ RSRL''. We build up the risk-sensitive distributional dynamic programming and use it to solve the Risk-sensitive MDP problem. This connection is more natural and useful.
> 2. Policy evaluation v.s. control. The claim of [22] is only valid when a policy is given. However, we establish the DDP that applies to policy evaluation (when a policy is given) and control (finding the optimal policy).
> 3. Estimate v.s. exact solution. It is impossible to obtain the true MGF of $Z^{\pi}_h(s, a)$ according to [22] since that requires solving infinitely many exponential Bellman equations. Hence [22] can only be used to estimate the MGF by choosing a grid of $\beta$s. In contrast, our framework exactly solves the distributional bellman equation.

---

> > ### Author Response · Authors · 2022-08-02
> > **Response to Reviewer SdUB(2)**
> >
> > **Q3** ''Does the improvement in the lower bound in Section 5.3 come from the newly proposed framework or just different proof techniques?''
> > **A**: The improvement in the lower bound is due to different proof techniques, not the distributional framework. The proposed lower bound is minimax/information-theoretic rather than the one for our DRL algorithm. In other words, it implies that *any* algorithm must suffer the claimed regret in the worst case. The proof of [22] reduces the regret lower bound to the two-armed bandit regret lower bound. Since the two-armed bandit is a special case of MDP with $S=1$, $A=2$ and $H=1$, the reduction-based proof only leads to a lower bound independent of $S, A$, and $H$. Instead, our tight lower bound follows a totally different roadmap, which is motivated by [19]. [19] proves the tight minimax lower bound $H\sqrt{SAT}$ for risk-neutral MDP. However, the generalization to risk-sensitive MDP with entropic risk measure is non-trivial. The main technical challenge is dealing with the non-linearity nature of the entropic risk measure. The proof in [19] heavily relies on the fact the expectation is linear, allowing the exchange between taking the risk measure (expectation/mean) and the summation. In the risk-sensitive setting, the non-linearity of ERM involves new proof techniques.

---

> ### Author Response · Authors · 2022-08-07
> **Upload of the Revised Paper**
>
> Dear reviewer SdUB,
>
> We have uploaded a revised version of the paper, in which the revised part is highlighted on the red front. We hope that the revised paper could better address your concern. Feel free to reply if you have any questions.

---

### Official Review · Reviewer_CsUD · 2022-07-13

**Rating:** 6
**Confidence:** 3
**Soundness:** 3 good
**Presentation:** 3 good
**Contribution:** 2 fair

**Summary:**

The paper deals with a risk-sensitive reinforcement learning problem where the risk functional is the entropic risk. The authors provide both the regret upper bounds for a proposed model-based algorithm and a model-free algorithm and a tighter regret lower bound compared to prior works.


**Questions:**

- The authors suggest that the lower bound proof in [22] has a mistake. Can the authors elaborate more on this?
- Can the authors compare their proposed algorithms with that proposed in [22] in a more detailed way to explain how optimism may have been used differently in this paper and what that implies?


----

Stylistic suggestions:

- Using ERM to refer to entropic risk measure may be confusing since ERM is known to be Empirical Risk Minimization in standard ML literature.
- \citep should be used for the first two citations in L101.
- In L168, (s,a)s may be confusing to the reader.


**Limitations:**

Yes.

**Strengths And Weaknesses:**

Strength: The ideas in the paper are clearly explained. A tighter lower bound is presented in the paper.

Weakness: Given that prior works, e.g., [22, 23], have also used optimism to solve the exact same problem and achieve similar regret guarantees, it is unclear how much novelty/significance the newly proposed algorithms are.

---

> ### Author Response · Authors · 2022-08-02
> **Response to Reviewer CsUD**
>
> Thank you for the constructive feedback. We would like to address your concerns/questions as follows.
> ***
> **W1** ''Given that prior works, e.g., [22, 23], have also used optimism to solve the exact same problem and achieve similar regret guarantees, it is unclear how much novelty/significance the newly proposed algorithms are''
> **A**: **Significance**. We give a summary of the most related prior works here. [22,23] solved the risk-sensitive MDP problem using *valued-based* RL, which estimates and constructs the optimistic version of the (ERM) value function. [22] proposed the RSVI2 algorithm that improved upon [23] and achieved the SOTA result with the regret upper  bound of $\tilde{\mathcal{O}}(\frac{\exp(|\beta| H)-1}{|\beta|}H\sqrt{S^2AK})$. The significance of the proposed algorithms is three-fold.
> 1. Our algorithms are the first distributional reinforcement learning algorithms with provably regret guarantees, suggesting that DRL can work well and even matches the performance of the SOTA value-based RL algorithm for risk-sensitive control in terms of sample complexity. The idea of leveraging the distributional information for risk-sensitivity purposes is natural since the risk measure value is obtained by applying the risk measure/functional to the return distribution. However, existing works on risk-sensitive control via DRL approaches [12, 31, 1] lack regret analysis. Thus, it is difficult to evaluate and improve their algorithms for sample efficiency. Therefore, our algorithms with near-optimal regret upper bounds bridge the gap between the DRL and risk-sensitive MDP in the theoretic RL community.
> 2. Compared with [21], our algorithms are simpler and easier to interpret, leading to clean regret analysis. [21] implements optimism by adding a bonus to the risk measure value function. It designed an exploration mechanism called doubly decaying bonus to remove the $\exp(|\beta| H^2)$ factor from [22]. The doubly decaying bonus decays across the episode and the horizon, which is complicated and not straightforward. Instead, our algorithms implement the distributional optimism by iteratively constructing the optimistic return distribution. The distributional optimism does not involve a complicated bonus design. It only requires a simple application of distributional optimism operator with a constant decaying across the episode. Moreover, the doubly decaying bonus obscures the regret analysis, while our distributional-based analysis is clean and easy to follow.
> 3. Our algorithm may be generalized to risk-sensitive MDP with other risk measures. The analysis of [22,23] is particularly suitable for the ERM. It is unclear whether it is possible to extend to other risk measures. Under the distributional perspective, our algorithm maintains a sequence of optimistically plausible estimates of the return distribution. Since the distributional information suffices to deal with any risk measure, our algorithm may motivate the design of similar algorithms for other risk measures.
> **Novelty**. The novelty of the algorithm is mainly the distributional optimism. This point is closely related to **Q2**, so we leave it to the response to **Q2**.
> ***
> **Q1** ''Can the authors elaborate more on the mistakes of the lower bound in [21]?''
> **A**:     We indeed provide a thorough description of that in Appendix C.3.1., in which we explain why the proof of their lower bound is incorrect, and correcting their proof leads to a weaker lower bound than they had claimed. However, the connection to Appendix C.3.1. is missing in the main text. We will add it to remind the readers.   The major mistake appears in the second inequality of the following statements in the proof [22],
>     \begin{align*}
> \mathbb{E}[\operatorname{Regret}(K)] &\gtrsim \frac{\exp(\beta H/2)-1}{\beta}\sqrt{K\log(K)}
> &\gtrsim \frac{\exp(\beta H/2)-1}{\beta}\sqrt{KH\log(KH)}.
> \end{align*}
> The authors establish the second inequality based on the following fact (Fact 5, [22]):
> 	For any $\alpha>0$, the function $f_{\alpha}:=\frac{e^{\alpha x}-1}{x}, x>0$ is increasing and satisfies $\lim_{x\rightarrow0}f_{\alpha}=\alpha$.
>     Nevertheless, we can only use this fact to derive $\frac{\exp(\beta H/2)-1}{\beta}\gtrsim H$, which combined with the first inequality yields \begin{align*}
>     \mathbb{E}[\operatorname{Regret}(K)] \gtrsim H\sqrt{KH\log(KH)}.
>     \end{align*}
> It is a weaker lower bound and does not show the dependence on $\beta$. The best result we can get based on the original proof of [22] is that
> 	$$
> 	\mathbb{E}[\operatorname{Regret}(K)] \gtrsim \frac{e^{|\beta| H / 2}-1}{|\beta|} \sqrt{K \log K}.
> 	$$
> 	For comparison, we provide a lower bound for $\beta>0$ as follows.
> 	$$
> 	\mathbb{E}[\text{Regret}(K)]\gtrsim\frac{\exp(\beta H/6)-1}{\beta H}H\sqrt{SAT}.
> 	$$

---

> > ### Author Response · Authors · 2022-08-02
> > **Response to Reviewer CsUD(2)**
> >
> > The corrected lower bound is worse than our bound with a factor of $\sqrt{SAH}$. It is difficult to improve the bound based on their original proof, which essentially reduces to a two-armed bandit lower bound. Since two-armed bandit is a special case of MDP with $S=1$, $A=2$ and $H=1$, the reduction-based proof only leads to a lower bound independent of $S$ and $H$. It is unclear how to construct a hard MDP instance with an arbitrary value of $S, A$ and $H$ from the hard two-armed bandit instance. Instead, our tight lower bound follows a different roadmap, which is motivated by [19]. [19] proves the tight minimax lower bound $H\sqrt{SAT}$ for risk-neutral MDP. However, the generalization to risk-sensitive MDP with entropic risk measure is non-trivial. The main technical challenge is dealing with the non-linearity nature of the entropic risk measure. The proof in [19] heavily relies on the fact the expectation is linear, allowing the exchange between taking the risk measure (expectation/mean) and the summation. In the entropic risk measure case, the non-linearity involves new proof techniques.
> > ***
> > **Q2** ''Can the authors compare their proposed algorithms with that proposed in [22] in a more detailed way to explain how optimism may have been used differently in this paper and what that implies''
> > **A**: Note that [22] solves the risk-sensitive MDP using a *value-based*  Bellman optimality equation for ERM
> > $$
> > Q_{h}^*(s, a) =r_{h}(s, a)+\frac{1}{\beta} \log  \mathbb{E} \left.[ \text{exp} (\beta \cdot V_{h+1}^* (S')) \right],
> > $$
> > $$
> > V_{h}^* (s) =\max_{a \in \mathcal{A}} Q_{h}^*(s, a),  V_{H+1}^* (s)=0,
> > $$
> > where the expectation in the first equality is taken w.r.t. $S' \sim P_h(\cdot |s,a)$. This equation presents an iterative way of directly computing the optimal ERM value function when the model is known. In contrast, the foundation of our algorithm is the *distributional* Bellman optimality equation (see equation (1)) that iteratively computes the optimal return distribution. In each iteration,  [22] first obtains a non-optimistic estimate of the value function using the Bellman optimality equation. It then adds a bonus term $b^k_h(s,a)$ to the estimated value function for each state-action pair to construct the optimistic value function, where the bonus term decays with the visitation counts $N_h^k(s,a)$.
> > Compared to the bonus-based optimism, our algorithms implement the *distributional optimism* that iteratively constructs the optimistic version of the return distribution. In particular, distributional optimism is enforced by the two types of distributional optimism operators (see Section 4.2). The distributional optimism operator takes the non-optimistic estimated return distribution as input and outputs the optimistic return distribution. The distributional optimism operator enables us to obtain the optimistically plausible return distribution over the confidence set around the empirical distribution. By the monotonicity of ERM, the more optimistic the distribution is, the larger its ERM value is.
> > Consequently, the optimistic return distribution w.r.t. the true optimal one leads to the optimistic ERM value. In contrast, the bonus-based optimism directly adds bonus terms to the estimates of value functions. Furthermore, the distributional optimism induces the distributional analysis. Our analysis is clean and can better handle the error propagation across time-steps, leading to a $\exp(|\beta| H^2)$ tighter bound than [22].
> > ***
> > **Stylistic suggestions**. We appreciate that you provide the stylistic suggestions. We will follow them in the revised version of the paper.

---

> ### Author Response · Authors · 2022-08-07
> **Upload of the Revised Paper**
>
> Dear reviewer CsUD,
>
> We have uploaded a revised version of the paper, in which the revised part is highlighted on the red front. We hope that the revised paper could better address your concern. Feel free to reply if you have any questions.

---

### Official Review · Reviewer_vdwj · 2022-07-16

**Rating:** 7
**Confidence:** 3
**Soundness:** 3 good
**Presentation:** 3 good
**Contribution:** 3 good

**Summary:**

This paper is about distributional reinforcement learning (DRL), which aims at learning optimal policies (in terms of total payoff) in a Markov decision process. While the classical reinforcement learning (RL) paradigm is based on the expected total return, in DRL one is interested about the whole probability distribution of this total return.

In this work, the idea is to leverage this distributional information for risk-sensitivity purpose.
The focus is on the entropic risk-measure (ERM) of the total reward in an episodic MDP. Properties of ERM are given (additivity and monotonicity preserving) leading to an optimality principle akin to the classical one based on expected value.
Two algorithms (one model-free and the other model-based) called RODI are proposed: they both rely on the OFU principle applied to ERM.
While most risk-aware DRL papers have no regret analysis, here both upper and lower regret bounds are provided.

**Questions:**

(Minor) questions/comments:

- line 80: more accurate to say that P_h maps state-action pair to probability distribution over S, i.e. P_h : SxA -> Delta(S)
- line 108: replace Z(s) by Y(s)
- footnote page 4: strange formulation? maybe rather say: "The algorithms for random reward enjoy regret bounds of the same order"
- notational issue: at line 176, reference to "c_h^k" but in Algorithm 1 it is just written "c_h". Same holds for eta_h. Is it normal or a notation issue?
- what is f_h in Proposition 2?

**Limitations:**

The scope of the paper is limited to the ERM criterion, but it is a reasonable "limitation" in order to design a risk-sensitive DRL approach.

**Strengths And Weaknesses:**

The paper is well-written and motivated by risk-sensitive applications. The DRL perspective makes a lot of sense to tackle risk-aware problems. The main strength of the paper is that its algorithmic RODI procedures are naturally derived from DRL applied to ERM and combined with the OFU principle.

The main weakness is the lack of numerical experiments, which is balanced by the theoretical nature of the paper.

---

> ### Author Response · Authors · 2022-08-02
> **Response to Reviewer vdwj**
>
> Thank you for the constructive feedback. We would like to address your concerns/questions as follows.
> ***
> **Q1** ''Line 80''.
> **A**: This is a good suggestion, and we will follow it in the revised version.
> ***
> **Q2&Q3** ''Line 108 and footnote page 4''
> **A**:Thank you for pointing out the typos. We will fix them in the revised version.
> ***
> **Q4** ''Notational issue at line 176''
> **A**: This is normal. While there is no superscript $k$ in $c_h$ and $\eta_h$ in Algorithm 1,  we introduce the superscript $k$ in the main text to distinguish the iterates in different episodes (see Line 167).
> ***
> **Q5** ''what is $f_h$ in Proposition 2?''
> **A**: $f_h(s,a)$ is the probability density function (PDF) of reward distribution  $R_h(s,a)$, which is missing in the context. We will add the definition of $f_h$ in the revised version. Now we provide some intuitions behind why $f_h$ appears in Prop.2. For two *independent* r.v.s $X$ and $Y$, let $F_X$, $F_Y$, $f_X$ and $f_Y$ denote their CDFs and PDFs, then the CDF of their sum $Z=X+Y$ is given by $F_Z(z) = (F_X * f_Y)(z)=\int_{\mathbb{R}}F_X(x) f_Y(z-x)dx$. Given that $Z_h^*(s,a)=R_h(s,a)+Y_{h+1}^*(s')$  where $R_h(s,a)$ is independent of $Y_{h+1}^*(s')$, since $R_h(s,a)\sim R_h(s,a)$ and $Y_{h+1}^*(s') \sim \left.[P_h \eta_{h+1}^* \right] (s,a)$, we have $Z_h^* (s,a)\sim \left.[P_h\eta_{h+1}^* \right] (s,a) * f_h(s,a)$.

---

> ### Author Response · Authors · 2022-08-07
> **Upload of the Revised Paper**
>
> Dear reviewer vdwj,
>
> We have uploaded a revised version of the paper, in which the revised part is highlighted on the red front. We hope that the revised paper could better address your concern. Feel free to reply if you have any questions.

---

### Official Review · Reviewer_wafk · 2022-07-18

**Rating:** 5
**Confidence:** 3
**Soundness:** 3 good
**Presentation:** 3 good
**Contribution:** 3 good

**Summary:**

This paper considers finite episodic Markov decision processes aiming with risk-sensitivity. Two properties are identified to develop both the model-based and model-free algorithms for risk-sensitive distributional dynamic programming.  Both upper and lower bounds for regret haven been proved.

**Questions:**

How does the performance of the proposed algorithms compare to that of DRL algorithms using CVaR? Is it possible to extend the proposed algorithms for continuous control tasks, such as Cartpole or more complicated robotic learning application? Can the authors clarify the unique novelty of the regret analysis technique?



**Limitations:**

Yes, the authors have discussed the limitations of their theory. However, I mean some study on the actual performance is necessary for this paper. The proposed regret guarantees are meaningful only when the algorithms actually do perform well.

**Strengths And Weaknesses:**

Strengths:
1. A specific entropic measure is used, enable the derivations of useful distributional Bellman optimality equation as well as risk-sensitive distributional dynamic programming. This contribution is very interesting.

2. OFU principle has been used to develop deep RL algorithms with provable regret bounds.

3. An improved lower bound has been obtained when beta is positive.

4. The paper is well written. The presentation is clear.


Weakness:
1. This paper focuses on the episodic MDP setting. The most common setting in the risk-sensitive control literature is the setting with continuous state/action spaces (see Jacobson 1973 or Whittle 1990). Then stability also becomes an important issue. It is unclear how useful the results in this paper will be for continuous control applications. Maybe rephrase and focus on "risk-sensitive MDP"?

2. This paper proposes two algorithms with provable regrets. However, it is unclear whether the proposed algorithms will really perform well on any MDP tasks. Numerical examples are needed in this case. This is different from analyzing existing algorithms which already perform well.

3. CVaR vs Entropic Risk Measure (ERM):  CVaR has been used a lot in deep RL tasks and seems to be a better risk measure than ERM for deep RL. One reason why ERM was popular in some early risk-sensitive control literature is that for linear dynamics, ERM leads to a dynamical programming solution with relatively simple analytical form. In the deep RL setting, it is unclear whether ERM has any practical advantages over CVaR.

4. The regret analysis technique is not that new (similar analysis has been used to study episodic MDPs (without ERM) before).

---

> ### Author Response · Authors · 2022-08-02
> **Response to Reviewer wafk**
>
> Thank you for the constructive feedback. We would like to address your concerns as follows. We first clarify a possible misunderstanding of the acronym ''DRL'' (line 20). Throughout the paper,  ''DRL'' standards for Distributional Reinforcement Learning instead of the well-known Deep Reinforcement Learning. While we introduce the acronym in the abstract (line 4), the full description of ''DRL'' is missing in the main text (line 20). We are sorry about causing the confusion, and we will fix it in the revised version.
> ***
> **W1/Q2**  ''It is unclear how useful the results in this paper will be for continuous control applications'' '' Maybe rephrase and focus on ''risk-sensitive MDP"''
> **A**: There is a convention in RL literature to use ''control'' to refer to finding the optimal policy. This is in contrast with the term ''policy evaluation'', which represents evaluating the value of a given policy. In this sense, ''control'' is orthogonal to the continuity of action space. This paper adopts the term ''risk-sensitive control'' since we aim to find the optimal policy under the risk-sensitive criteria. This paper focus on the tabular MDP setting, in which the state space and action space are both finite. It is a standard setting for regret analysis of RL (see also [Azar et al. (2017); Jin et al. (2018)]). In this work, tabular MDP serves as the first attempt to bridge the gap between the distributional RL methods and the risk-sensitive community. It will be a meaningful direction to extend the results to the function approximation (continuous control) setting, which is left as future work (see also line 298).
> ***
> **W2** ''Numerical examples are needed ''
> **A**: This work focus on the sample complexity of distributional reinforcement learning applied to risk-sensitive RL. We aim to verify the effectiveness of distributional reinforcement learning for a risk-sensitive purpose. Some existing works use distributional reinforcement learning approaches to solve risk-sensitive tasks, but all lack regret analysis. It is thus challenging to evaluate and improve the algorithms for sample efficiency. This work is the first one to bridge the gap between distributional RL and Risk-sensitive RL in the aspects of sample complexity. Admittedly, it is an excellent suggestion to conduct numerical experiments to support our theoretical results. We will work on this in the future.
> ***
> **W3/Q1** ''In the deep RL setting, it is unclear whether ERM has any practical advantages over CVaR''  ''How does the performance of the proposed algorithms compare to that of DRL algorithms using CVaR''
> **A**: Maybe you are confused with our setting here. We focus on the episodic and tabular MDP in this paper, which is a standard setup for studying the *sample complexity* of RL algorithms. Under this setting, only a few works [22,23]  take the risk sensitivity into account. They all assume that the risk measure is the entropic risk measure. Meanwhile, it is still unclear whether efficient algorithms exist for risk-sensitive episodic MDP, with CVaR being the risk measure. The major challenge of CVaR is that the optimal policy for CVaR may be non-Markovian/history-dependent since the dynamic programming does not hold. It is an interesting direction to explore CVaR in the episodic MDP setting.
> ***
> **W4/Q3** ''Can the authors clarify the unique novelty of the regret analysis technique?''
> **A**: Our regret analysis is novel, and the major novelty is demonstrated in the form of *technical highlights*. Compared to the traditional/non-distributional analysis, which works with one-dimensional values, our analysis is distribution-centered, called the *distributional analysis*. The distributional analysis deals with the distributions of the return rather than the risk measure values of the return. For example, it involves the operations of the distributions, the optimism between different distributions, the error caused by estimation of distribution, etc. These distributional aspects fundamentally differ from the traditional analysis that deals with the one-dimensional scalars (value functions). Now we recap the novelty in the following.

---

> > ### Author Response · Authors · 2022-08-02
> > **Reponse to Reviewer wafk(2)**
> >
> > 1.  **Property of the risk measure**. We identify two crucial properties of EERM that establish the regret upper bounds, including the Lipschitz continuity and linearity. These two properties were not discovered and used in the previous analysis. The Lipschitz continuity property enables the regret upper bounds of DRL algorithms since it provides a way to bound the error in ERM value caused by the error in the estimation of distributions. The linearity further tightens the bound by a factor of $\exp(\beta H^2)$  due to better control of error back-propagation across time-steps.
> > 2.  **Distributional optimism**. The traditional analysis uses optimism in the face of uncertainty to construct a sequence of optimistic value functions. However, our analysis implements the  ''distributional optimism'' that yields a sequence of optimistic return distributions. In particular, we use the concentration inequalities of distributions together with the ''key properties'' of ERM, and these techniques are novel.

---

> ### Author Response · Authors · 2022-08-07
> **Upload of the Revised Paper**
>
> Dear reviewer wafk,
>
> We have uploaded a revised version of the paper, in which the revised part is highlighted on the red front. We hope that the revised paper could better address your concern. Feel free to reply if you have any questions.

---

### Meta-Review · Area_Chair_Cndw · 2022-08-27

**Recommendation:** Reject
**Confidence:** Certain

**Metareview:**

While this work provides interesting insights on  distributional reinforcement learning for risk-sensitive control, it is unclear how beneficial these results are given the closely related work [22] (on the same problem with the same regret bound). The authors mentioned in the rebuttal that their algorithm may motivate the design of similar algorithms for other risk measures, but no concrete discussions and examples were provided. We believe that the paper would benefit from another round of revision to properly address these issues and make its contributions move convincing.

**Award:**

No

---

### Decision · Program_Chairs · 2022-09-14

Reject